# Annual vegetation maps in Qinghai-Tibet Plateau (QTP) from 2000 to 2022 based on MODIS series satellite imagery

Guangsheng Zhou[1†], Hongrui Ren[2†], Lei Zhang[2], Xiaomin Lv[1], and Mengzi Zhou[1]

[1]State Key Laboratory of Severe Weather, Chinese Academy of Meteorological Sciences, Beijing 100081, China
[2]Department of Geomatics, Taiyuan University of Technology, Taiyuan 030024, China

*Correspondence*: Guangsheng Zhou (zhougs@cma.gov.cn)

† Co-first authors

**Abstract.** The Qinghai Tibet Plateau (QTP), known as the "Third Pole of the Earth" and the "Water Tower of Asia", plays a crucial role in global climate regulation, biodiversity conservation, and regional socio-economic development. Continuous annual vegetation types and their geographical distribution data are essential for studying the response and adaptation of vegetation to climate change. However, there is very limited data on vegetation types and their geographical distributions on the QTP due to harsh natural environment. Currently, land cover/surface vegetation (LCSV) data are typically obtained using traditional classification methods for each period's product, based on remote sensing information. These approaches do not consider the temporal continuity of vegetation presence, leading to a gradual increase in misclassified pixels and uncertainty in their locations, consequently decreasing the interpretability of the long-time series remote sensing products. To address this issue, this study developed a new method for long-time continuous annual vegetation mapping based on reference vegetation maps and annual updates, and mapped the vegetation of the QTP from 2000 to 2022 at a 500 m spatial resolution through the MOD09A1 product. The overall accuracy of continuous annual QTP vegetation mapping from 2000 to 2022 reached 83.27%, with the reference annual 2020 reaching an accuracy of 83.32% and a Kappa coefficient of 0.82. This study supports the use of remote sensing data for long-term continuous annual vegetation mapping.

## 1 Introduction

Vegetation, an integral component of Earth's ecosystems, plays an irreplaceable role in maintaining climate stability, preserving biodiversity, and supplying vital resources for humans. Vegetation maps not only facilitate a visual comprehension of vegetation types and their geographic distribution, but also provide essential data for natural resource management and environmental protection (Immerzeel et al., 2010). Particularly, long-time series of geographical distribution data on vegetation types are crucial for revealing the impacts of climate change and human activities on vegetation, elucidating the succession processes of vegetation, and scientifically managing vegetation ecosystems.

The availability of large-scale, long-term, and free remote sensing imagery has significantly advanced the development of land cover and surface vegetation (LCSV) data. Notable international products include the European Space Agency (ESA)'s

GlobCover_2005 and GlobCover_2009 (300 m) (Bontemps et al., 2010), as well as the ESA Climate Change Initiative (ESA-CCI) land cover dataset, which offers improved temporal continuity at a 300 m spatial resolution (Defourny et al., 2017). Additionally, the National Aeronautics and Space Administration (NASA)'s annual MCD12Q1 product since 2001 (500 m) (Friedl et al., 2010; Sulla-Menashe et al., 2019) is widely used. In China, representative products include GlobeLand30 by Chen et al. (2015), Finer Resolution Observation and Monitoring of Global Land Cover (FROM_GLC)10 and FROM_GLC30

by Tsinghua University (Gong et al., 2013; Gong et al., 2019), Global 30m Land-cover Classification with a Fine Classification System (GLC_FCS30) and Global 30m Land-cover Dynamics Monitoring Dataset (GLC_FCS30D) by the Chinese Academy of Sciences (Zhang et al., 2019c; Zhang et al., 2021b; Zhang et al., 2024b), and China Land Cover Dataset (CLCD) by Wuhan University (Yang and Huang, 2021). Leveraging the intelligent remote sensing mapping (iMap) concept and framework, there has been rapid progress in developing global-scale seamless daily data cubes and in creating annual and seasonal land surface

maps (Feng and Li, 2020; Liu et al., 2021a).

Traditional long-time series LCSV datasets are often generated by independently classifying each period's product. However, significant spatial distribution differences between products from different periods hinder their direct comparability (Liu et al., 2021b). For instance, NASA's MCD12Q1.v5 product, produced using MODIS data and a decision tree classifier (Friedl et al., 2010), exhibited substantial instability, with an annual land cover label change rate of approximately 11.4%

between 2001 and 2013. Similarly, GlobCover_2009 and GlobCover_2005 show notable spatial distribution differences, making them unsuitable for change detection studies (Bontemps et al., 2010). To address this issue, post-processing methods have been developed and have demonstrated remarkable effectiveness. For example, the Hidden Markov Model (HMM) method (Sulla-Menashe et al., 2019), applied to the MCD12Q1.v6 product, reduced the instability of land cover label changes to 1.6%, compared to MCD12Q1.v5. Another study, using spatial-temporal consistency methods (Yang and Huang, 2021),

further improved the mapping accuracy of land cover time series datasets. However, these post-processing methods rely on the establishment of numerous subjective rules, leading to complexity and limitations in their application (Zhang et al., 2024b). In recent years, research on continuous and dense change detection methods has made rapid progress, particularly with the Continuous Change Detection and Classification (CCDC) algorithm (Zhu and Woodcock, 2014; Zhu et al., 2019). This technique leverages all available remote sensing data to conduct continuous land cover change detection, providing a new

approach to address the aforementioned issues. For instance, Xian et al. (2022) used the CCDC method to produce annual land cover products for the United States from 1985 to 2017, while Zhang et al. (2024b) applied it to generate the global GLC-FCS30D product, which demonstrates greater temporal stability compared to GLC-FCS30. Therefore, there is an urgent need to develop long-time series mapping methods based on change detection to overcome the limitations of traditional classification and post-processing approaches.

The QTP, known as the "Roof of the World" and the "Water Tower of Asia," is critical for global climate regulation and regional socio-economic development (Yao et al., 2012). In the 1970s, China carried out the first extensive scientific survey of the QTP. Currently, the second QTP scientific expedition is underway, with vegetation survey being one of the major components (Zhou et al., 2023). Understanding the evolution of vegetation types on the QTP is important for revealing the

effects of climate change on vegetation structure and function. Additionally, such insights are essential for elucidating the

carbon and water cycles of the QTP and for formulating high-quality, sustainable development strategies for the region amidst global warming (Wang et al., 2022; Wang et al., 2023; Zhang et al., 2024a).

As the "Third Pole" of the Earth, the QTP has very limited data on vegetation types and geographical distribution due to its harsh natural environment. Currently, the primary vegetation data for the QTP include the "Vegetation map of Qinghai Tibet Plateau in 2020 with 10 m spatial resolution" (Zhou et al., 2022a), the "A new vegetation map for Qinghai-Tibet Plateau by

integrated classification from multi-source data products (2020)" (Zhang et al., 2021a), the "Vegetation map of Qinghai Tibet Plateau in 1980s" (Zhou et al., 2022b), and the "Vegetation map of Qinghai-Tibet Plateau permafrost zone" (Wang et al., 2016). These vegetation distribution products are single-period maps, which are insufficient to depict dynamic changes in vegetation, thus limiting the understanding of vegetation evolution trends and mechanisms on the QTP. In addition, Wang et al. (2022) described the distribution of alpine meadows and alpine steppes on the QTP and predicted the distribution trends of these

vegetation types over the next century using precipitation data. Wang et al. (2023) analyzed the characteristics of grassland changes on the QTP over the past 40 years from the perspective of plant community structure. Other studies have focused on wetlands, water bodies, glaciers, and other aspects of the QTP (Zhang et al., 2019a; Hu et al., 2023; Li et al., 2023). However, these long-time series products, created with traditional classification methods, are primarily focused on specific types, overlooking the need to identify temporal changes across different types, which limits the accuracy of the products. Therefore,

in order to provide continuous data support for research on the interaction between vegetation and climate change, there is an urgent need for long-time series vegetation mapping.

This study aims to develop a new approach to long-time series vegetation mapping using remote sensing imagery, and to map vegetation of the entire QTP at 500 m from 2000 to 2022, using the MOD09A1 remote sensing data.

## 2 Materials and Methods

### 2.1 Study Area

The QTP, situated between 25°59′30″ N and 40°1′0″ N, 67°40′37″ E and 104°40′57″ E, has an average elevation of about 4320 m and a total area of 3.08 million km² (Zhang et al., 2021c). This study covers the entire QTP, spanning six provinces and regions within China - Tibet, Qinghai, Gansu, Sichuan, Yunnan, and Xinjiang - as well as areas in India, Pakistan, Tajikistan, Afghanistan, Nepal, Bhutan, Myanmar, and Kyrgyzstan. In particular, the area of the QTP within China is about

2.58 million km² (approximately 83.7% of the QTP) with an average elevation of about 4400 m (Yu et al., 2014; Zhang, 2019; Zhang et al., 2021d; Zhang et al., 2021e). The QTP slopes downward from the high northwest to the lower southeast, with a humid and rainy climate in the southeast and arid conditions in the northwest. The vegetation distribution across the QTP is influenced by topography and climatic conditions, revealing a clear horizontal zonation ranging from forests, shrubs, and meadows in the southeast to grasslands and deserts in the northwest. Additionally, as temperatures decrease with increasing

altitude, there is a distinct vertical stratification in vegetation, ranging from forests at lower elevations to alpine meadows and alpine vegetation at higher elevations.

## 2.2 Data Sources

### 2.2.1 Training Sample Data for Vegetation Type

The vegetation of the QTP is primarily categorized into 15 types (Editorial Board of the Vegetation Map of China, Chinese Academy of Sciences, 2007; Zhou et al., 2023). Given the significance of glaciers and snow cover, this study expanded the vegetation classification to 16 types for the 500 m spatial resolution mapping (Table 1), including evergreen broad-leaved forest (EBF), evergreen coniferous forest (ECF), coniferous and broad-leaved mixed forest (CBMF), deciduous broad-leaved forest (DBF), deciduous coniferous forest (DCF), scrub (SC), alpine scrub meadow (ASM), alpine meadow (AM), alpine grassland (AG), alpine vegetation (AV), alpine desert (AD), cultivated vegetation (CV), wetland (WE), water (WA), non-vegetated area (NVA), and glacier and snow (GS).

**Table 1. Vegetation classification system and training and validation sample counts on the QTP.**

| Types | Definitions | Training Samples | Validation Samples |
|---|---|---|---|
| Evergreen broad-leaved forest (EBF) | Forest communities composed of evergreen broad-leaved tree species | 936 | 76 |
| Evergreen coniferous forest (ECF) | Forest communities composed of evergreen coniferous tree species | 768 | 102 |
| Coniferous and broad-leaved mixed forest (CBMF) | Forest communities composed of both coniferous and broad-leaved tree species | 475 | 25 |
| Deciduous broad-leaved forest (DBF) | Forest communities composed of broad-leaved tree species that shed leaves in winter and grow in summer | 570 | 62 |
| Deciduous coniferous forest (DCF) | Forest communities composed of coniferous tree species that shed leaves in winter and grow in summer | 370 | 91 |
| Scrub (SC) | Vegetation communities dominated by shrubs | 397 | 79 |
| Alpine scrub meadow (ASM) | Vegetation communities composed of alpine shrubs and alpine meadows, adapted to cold, windy, dry, and alpine climates | 213 | 49 |
| Alpine meadow (AM) | Vegetation communities primarily consisting of herbaceous plants suited to cold climates | 1375 | 310 |
| Alpine grassland (AG) | Vegetation communities composed of alpine grasslands with low biomass and a short growing season | 872 | 217 |
| Alpine vegetation (AV) | Vegetation communities occurring above the treeline or shrub belt and below the permanent snowline, dominated by ice- and cold-tolerant plants | 503 | 97 |
| Alpine desert (AD) | Deserts composed of cold- and drought-tolerant cushion subshrubs | 499 | 325 |

| | | | |
|---|---|---|---|
| Cultivated vegetation (CV) | Vegetation communities formed through human cultivation | 469 | 109 |
| Wetland (WE) | Vegetation communities composed of water, bare soil, and herbaceous or woody plants | 246 | 81 |
| Water (WA) | Areas covered by water year-round | 377 | 91 |
| Non-vegetated area (NVA) | Areas composed of natural soil, sand, or rock | 204 | 36 |
| Glacier and snow (GS) | Areas covered by ice and snow year-round | 663 | 439 |

The training vegetation samples for the QTP within China were obtained from the "Vegetation map of Qinghai Tibet Plateau in 2020 with 10 m spatial resolution" (Zhou et al., 2022a). This product was created using a regional vegetation mapping method based on terrain-climate-remote sensing information, with a spatial resolution of 10 m and an overall accuracy of 89.5%. Its classification system is consistent with that of this study, except for the exclusion of GS, and was therefore used to generate the vegetation type training sample data for this research (Zhou et al., 2023). The vegetation map was resampled to 500 m. Subsequently, the dominant vegetation type and its proportional area, derived from the corresponding 50×50 pixels at 10 m spatial resolution, were determined as the vegetation type categorization and pixel purity (reflected as the percentage of the dominant vegetation type within these 50×50 pixels) for the 500 m spatial resolution map. Concurrently, different purity levels and their area proportions were calculated (Table 2). Random sampling was then conducted in areas with purity levels exceeding 70%, representing approximately 62.34% of the QTP. For the QTP regions outside China, vegetation type sample data were obtained from high-resolution Google Earth imagery from 2020. By visually interpreting these images, samples were selected from areas with consistent vegetation types within a 500 m range. Using these methods, a total of 8,937 training samples were collected to train the 2020 vegetation classification model (Table 1).

**Table 2. Purity and area proportion of 500 m resolution pixel samples of vegetation types on the QTP within China.**

| Purity (%) | 0-10 | 10-20 | 20-30 | 30-40 | 40-50 | 50-60 | 60-70 | 70-80 | 80-90 | 90-100 |
|---|---|---|---|---|---|---|---|---|---|---|
| Area Proportion (%) | 0.02 | 0.04 | 1.01 | 4.20 | 8.28 | 12.55 | 11.56 | 11.40 | 12.55 | 38.39 |

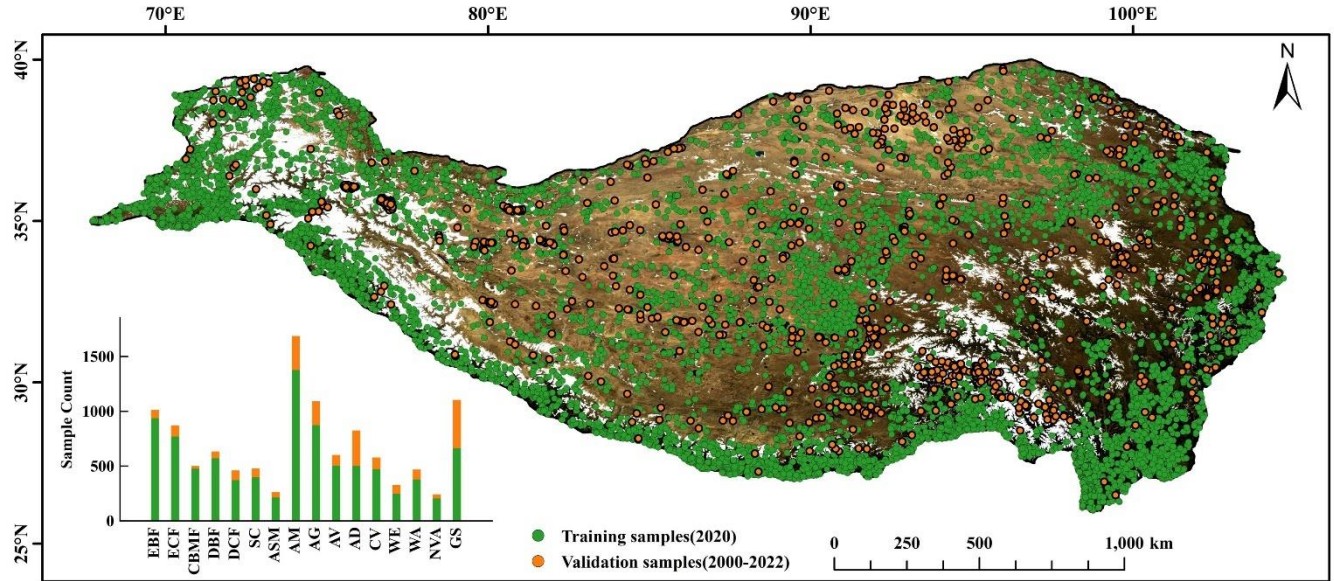

**Figure 1. Spatial distribution of training (2020) and validation (2000-2022) samples for vegetation mapping at 500 m resolution on the QTP.**

### 2.2.2 Validation Sample Data for Vegetation Type

As the "Third Pole of the Earth ", the QTP has high altitudes and sparse human activity, making it difficult to obtain validation samples. The validation samples in this study consist of three parts: field measurements, visual interpretation, and third-party samples.

The field measurement samples include 173 samples collected during field expeditions on the QTP in 2019, 2020, and 2021. These samples were collected during the vegetation growing season on the QTP, with neighboring samples spaced

approximately 50 km apart. Each sample is ensured to have uniform and consistent vegetation distribution within a 1 km×1 km area.

The visual interpretation samples (1002 in total) were obtained from previous work by Wu et al. (2024) using Google Earth and Google Earth Engine (GEE). The selected samples are from areas with consistent vegetation distribution from 1990 to 2020, and each sample covers an area greater than 500 m. Each sample was independently interpreted by three interpreters

using long-time series Landsat remote sensing images. Conflicting results were resolved through a second interpretation, and inconsistent samples were removed.

The third-party samples include the First All-season Sample Set (Li et al., 2017), the Global Land Cover Validation Samples (Zhang et al., 2021b), and validation samples gathered through literature search. The First All-season Sample Set consists of approximately 140,000 validation samples worldwide, obtained using Landsat 8 data from 2013-2015. The second set includes

44,043 validation samples, primarily from around 2015, and was published as a validation dataset for the GLC-FCS30-2015 product. The literature search samples were obtained from the China National Knowledge Infrastructure (https://www.cnki.net/)

and Web of Science (https://www.webofscience.com/) databases using keywords such as "Qinghai-Tibet Plateau," "vegetation", and "vegetation cover" to collect published validation samples from 2000 to 2022.

The third-party validation samples in this study carry certain uncertainties, such as spatial range and homogeneity. According to Li et al. (2017), the First All-season Sample Set includes samples with varying unit sizes (30 m, 100 m, 250 m, 500 m, and 1000 m), with those larger than 500 m being rare (only 10.6%). Samples smaller than 500 m are not suitable for validating 500 m vegetation maps. Additionally, the Global Land Cover Validation Samples from Zhang et al. (2021b) and those from literature searches only guarantee accuracy in sampling time and type, but not spatial extent. To address these issues, this study applied the method of Feng et al. (2012) to extract spatially homogeneous regions from MOD09A1 data by analyzing the range of values in a 3×3 grid of 500 m pixels. Thresholds for homogeneity were set for each band: 0.03 for blue, green, and red; 0.06 for near-infrared; and 0.03 for both shortwave infrared 1 and 2. If the range of values across all bands fell within these thresholds, the area was considered homogeneous. GS on the QTP are mostly classified as heterogeneous areas. Therefore, the samples labeled as GS in the third-party samples are visually interpreted to remove unreasonable ones. After applying this quality control, 1,014 third-party validation samples were obtained, including 327 from the global all-season sample library, 499 from the GLC-FCS30-2015 global validation set, and 188 from literature searches.

The validation sample dataset for this product contains 2,189 samples from field measurements, visual interpretation, and third-party samples (Table 1). Among them, 173 field measurement samples are available for validating the 2020 product There are 1,002 samples from visual interpretation, characterized by stable attributes over multiple years, making them suitable for validating annual products from 2000 to 2020. Furthermore, 1,014 third-party validation samples can be used to validate annual products from 2000 to 2019.

### 2.2.3 Remote Sensing Data

Remote sensing data in this study were sourced from the Moderate Resolution Imaging Spectroradiometer (MODIS) series satellite imagery provided by NASA. As one of the longest-operating Earth observation instruments currently in orbit, MODIS is a multi-spectral observation sensor launched in May 1999, enabling efficient and comprehensive observation of the Earth. The MOD09A1 surface reflectance dataset, obtained through the GEE platform (Gorelick et al., 2017), is a Level 3 product generated from daily surface reflectance observations using an 8-day compositing algorithm. This algorithm selects the best observation during the 8-day period based on criteria such as minimal cloud cover and highest observation quality (Vermote et al., 2015). The dataset has undergone standardized preprocessing provided by NASA, including radiometric calibration, atmospheric correction, and cloud masking based on quality assurance (QA) flags (Vermote et al., 2015; Vermote, 2021). To ensure the data were suitable for this study, additional processing steps were applied. The original sinusoidal projection of MOD09A1 was converted to the WGS84 geographic coordinate grid to maintain spatial consistency with other datasets, and the data were spatially subsetted to the QTP region to match the study area. The MOD09A1 dataset provides surface reflectance in seven spectral bands (Table 2) with a spatial resolution of 500 m, and 1,051 MOD09A1 images from January 1, 2000, to

December 31, 2022, were used to generate annual vegetation maps at a 500 m resolution for the QTP. These preprocessing
steps ensure the reliability and quality of the data for vegetation classification and long-term analysis.

### 2.2.4 Climate and Terrain Data

Temperature and precipitation are key driving factors for the distribution and dynamic changes of vegetation types on the QTP, and the incorporation of climate data significantly enhances the understanding of vegetation distribution on the QTP (Wang et al., 2022; Zhou et al., 2023; Zhang et al., 2024a). This study utilized climate data which included annual precipitation (AP)
and annual average temperature (AT) across the entire QTP from 2000 to 2022. For the QTP within China, climate data at 1,000 m were obtained from the National Tibetan Plateau Data Center, specifically the "1-km monthly precipitation dataset for China (1901-2023)" and the "1-km monthly mean temperature dataset for China (1901-2023)" (Peng et al., 2017a; Peng et al., 2017b; Peng, 2019; Peng et al., 2019; Ding and Peng, 2020; Peng, 2020). To derive the precipitation and temperature features required for this study, the monthly precipitation data were summed to calculate annual totals, while the monthly mean
temperature data were averaged over the year. In contrast, climate data for areas of the QTP outside China were derived from the CRU high-resolution gridded dataset, featuring a spatial resolution of approximately 50,000 m. The Digital Elevation Model (DEM) and derived features are crucial for vegetation mapping. The terrain data was from the SRTM by the USGS (Farr et al., 2007), with a spatial resolution of 30 m. At last, this study applied the mean sampling method in GEE to resample data on AP, AT, elevation, slope, and aspect variables derived from SRTM data to a 500 m spatial resolution for integration
into vegetation mapping.

### 2.3 Vegetation Mapping of the QTP at 500 m Resolution in 2020

### 2.3.1 Classification Platform and Algorithm

The GEE cloud platform offers a variety of machine learning models, such as SVM and RF, the latter of which was utilized in this study for vegetation mapping (Gorelick et al., 2017; Zhang et al., 2023). The RF model is an ensemble learning-based
machine learning algorithm that mitigates the overfitting risk inherent in single decision trees by constructing and integrating multiple decision trees (Breiman, 2001). This approach not only enhances the accuracy and stability of predictions but has also been widely applied in the vegetation mapping.

### 2.3.2 Construction of Vegetation Mapping Features

The features used in vegetation mapping are divided into four categories (Table 3): terrain (elevation, slope, aspect), climate
(AT and AP), surface reflectance (R, N, B, G, M, S1, S2), and 14 index features, which are constructed from the single-band surface reflectance. These features were derived from the MOD09A1 remote sensing imagery data spanning from January 1, 2020, to December 31, 2020. Additionally, six percentiles—15%, 30%, 45%, 60%, 75%, and 90%—were calculated for the 7 reflectance bands and the 14 indices, representing the time series characteristics of each pixel. The 15% and 90% percentiles

were used as substitutes for the minimum and maximum values of the time series observations, effectively mitigating the influence of extreme values in time-series data (Zhang et al., 2021b). The 30%, 45%, 60%, and 75% percentile features were selected to capture temporal trends uniformly while avoiding feature redundancy (Sulla-Menashe et al., 2019; Zhang et al., 2024b). A total of 131 features were formulated from these 4 categories for vegetation mapping on the QTP.

**Table 3. Vegetation mapping features at 500 m resolution on the QTP.**

| Category | Features | Formula | Description |
|---|---|---|---|
| Terrain | Elevation | | The height of the terrain above sea level (m) |
| | Slope | | The steepness of the terrain (°) |
| | Aspect | | The direction the terrain slope faces (°) |
| Climate | AT | | Annual average temperature |
| | AP | | Annual precipitation |
| Surface reflectance | R | Red | 620 – 670 nm |
| | N | Near-infrared | 841 – 876 nm |
| | B | Blue | 459 – 479 nm |
| | G | Green | 545 – 565 nm |
| | M | Mid-infrared | 1230 – 1250 nm |
| | S1 | Shortwave Infrared 1 | 1628 – 1652 nm |
| | S2 | Shortwave Infrared 2 | 2105 – 2155 nm |
| Vegetation Index | NDVI | $\dfrac{N-R}{N+R}$ | Normalized Difference Vegetation Index (Tucker, 1979) |
| | EVI | $2.5\dfrac{N-R}{N+6R-7.5B+1}$ | Enhanced Vegetation Index (Huete et al., 1999) |
| | RVI | $\dfrac{N}{R}$ | Ratio Vegetation Index (Crippen, 1990) |
| | DVI | $N-R$ | Difference Vegetation Index (Roujean and Breon, 1995) |
| | SAVI | $\dfrac{(N-R)*1.5}{N+R+0.5}$ | Soil Adjusted Vegetation Index (Huete, 1988) |
| | GCVI | $\dfrac{N}{G}-1$ | Green Chlorophyll Vegetation Index (Lobell et al., 2015) |
| | NIRV | $\dfrac{(N-R)*N}{N+R}$ | Near-Infrared Reflectance of Vegetation (Badgley et al., 2017) |
| Urban Index | NDBI | $\dfrac{S1-N}{S1+N}$ | Normalized Difference Built-up Index (Zha et al., 2003) |
| | IBI | $\dfrac{NDBI-(SAVI+(G-S1)/(G+S1))/2}{NDBI+(SAVI+(G-S1)/(G+S1))/2}$ | Index-based Built-up Index (Xu, 2008) |
| Water Index | NDWI | $\dfrac{G-N}{G+N}$ | Normalized Difference Water Index (McFeeters, 1996) |

| | LSWI | $\dfrac{N - S1}{N + S1}$ | Land Surface Water Index (Xiao et al., 2004) |
|---|---|---|---|
| Snow Index | NDSI | $\dfrac{G - S1}{G + S1}$ | Normalized Difference Snow Index (Hall et al., 1995) |
| | NDGlaI | $\dfrac{G - R}{G + R}$ | Normalized Difference Glacier Index (Keshri et al., 2009) |
| Soil Index | BI | $\dfrac{(S1 + R) - (N + B)}{(S1 + R) + (N + B)}$ | Bare Soil Index (Chen et al., 2004) |

### 2.3.3 Feature Importance Evaluation and Feature Selection

Among the 131 features used for vegetation mapping, 126 optical remote sensing features (excluding topography and climate) were constructed based on surface reflectance. These features are prone to severe collinearity issues, which can lead to model overfitting, increased computational costs, and diminished interpretability. To mitigate the issue of high collinearity among these features, the Variance Inflation Factor (VIF) was employed, providing the ratio of variance in a model with multicollinearity among features to the variance in a model where multicollinearity is absent (James et al., 2013; Ngabire et

al., 2022):

$$VIF_j = \frac{1}{1 - R_j^2} \tag{1}$$

where $VIF_j$ is the VIF for feature j, and $R_j^2$ is the squared multiple correlation coefficient obtained from the regression of feature j with all other features. A higher VIF value indicates more severe collinearity, with a $VIF_j$ greater than 30 suggesting significant collinearity in feature j. This study utilized the RF classifier to determine the optimal feature combination for vegetation classification. The importance of terrain, climate, and optical remote sensing features was assessed through RF

classifier and filtered using the VIF method (Ramosaj and Pauly, 2019, Zhang et al., 2019b). The features were then ranked according to the calculated importance, and the top-ranking feature, along with combinations such as the top two, top three, etc., were used to construct various RF models for all possible feature combinations. The feature set with the smallest out-of-bag error was chosen as the best feature set for vegetation mapping on the QTP.

### 2.3.4 Evaluation of Mapping Accuracy

An optimal combination of terrain, climate, and optical remote sensing data in 2020 was integrated to achieve vegetation mapping of the QTP based on RF model in this study. The mapping accuracy was evaluated using the confusion matrix method, which involved calculating the OA (Eq. 2), Kappa (Eq. 3), PA (Eq. 4), and UA (Eq. 5).

$$OA = \frac{\sum_{i=1}^{n} m_i}{N} \tag{2}$$

$$Kappa = \frac{N \times \sum_{i=1}^{n} m_i - \sum_{i=1}^{n}(G_i \times C_i)}{N^2 - \sum_{i=1}^{n}(G_i \times C_i)} \tag{3}$$

$$PA = \frac{m_i}{G_i} \tag{4}$$

$$UA = \frac{m_i}{C_i} \tag{5}$$

where $m_i$ is the count of correctly classified pixels for category i; n is the count of categories; N is the overall quantity of classified pixels; $C_i$ and $G_i$ are the total counts of pixels classified as and actually in category i, respectively; OA is the overall accuracy; Kappa is the Kappa coefficient; PA is the mapping accuracy; and UA is the user accuracy.

## 2.4 Continuous Annual Vegetation Mapping at 500 m Resolution from 2000 to 2022

Utilizing the 2020 vegetation map at a 500 m spatial resolution as the reference, this study dynamically updated the 2019 vegetation map of the QTP with various algorithms: continuous change detection (CCD) (Eq. 6), potential vegetation change area identification (Eq. 7), and actual vegetation type identification (Eq. 8-9). The process was then repeated, taking the newly updated 2019 map as the reference to dynamically update the 2018 vegetation map. This method was consistently employed to enable the dynamic updating annual vegetation maps from 2000 to 2019, as well as for 2021 and 2022.

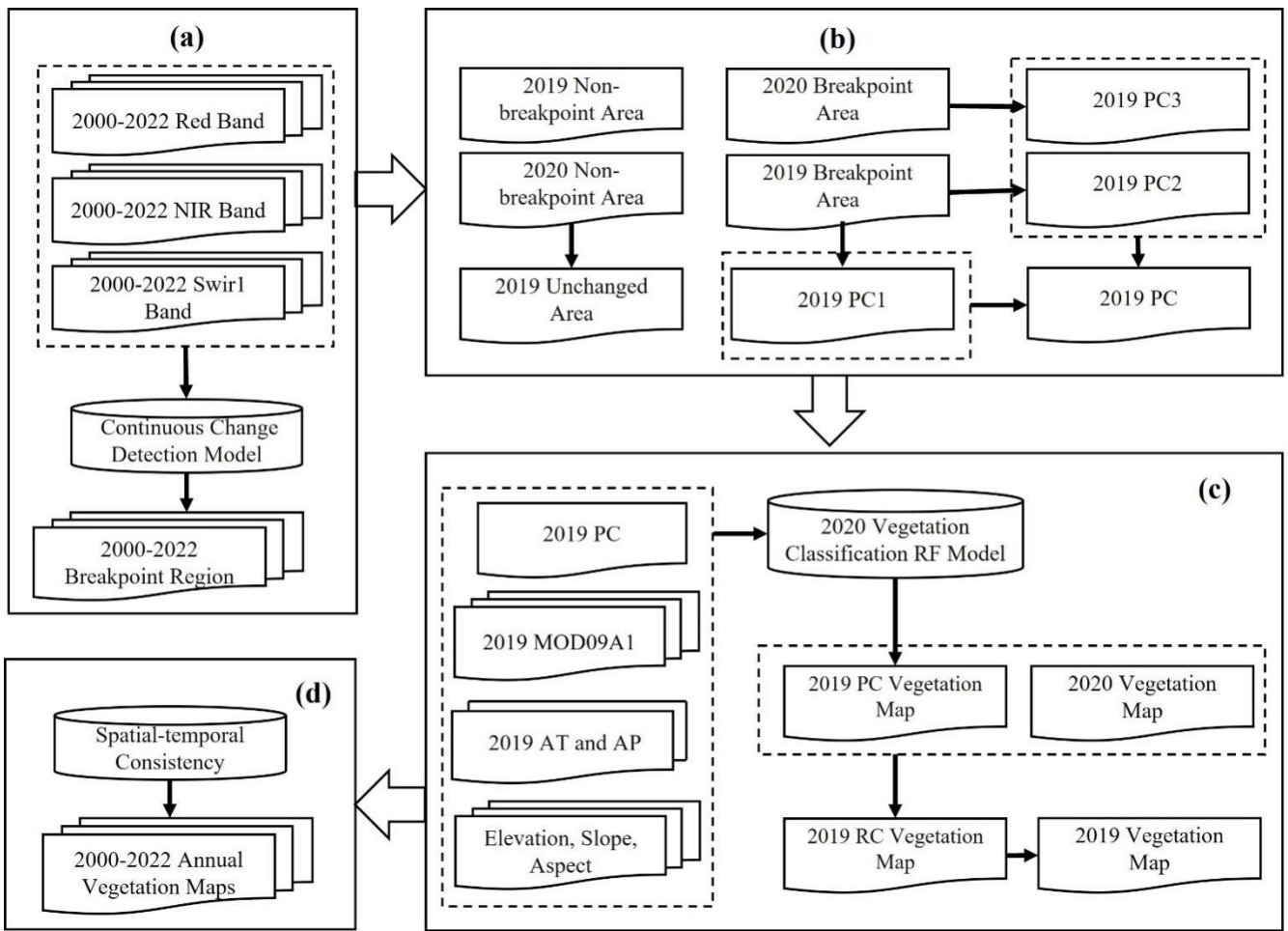

**Figure 2. Vegetation mapping technology route at 500 m resolution on the QTP from 2000 to 2022 (dynamically updating the 2019 vegetation map with the 2020 vegetation map as a reference).**

### 2.4.1 Continuous Change Detection Algorithm

The Continuous Change Detection and Classification (CCDC) algorithm, which simultaneously considers time series and spectral characteristics, is capable of accurately identifying temporal "breakpoints" in long-term remote sensing data. It has been applied in this study for long-term vegetation mapping. By dynamically detecting vegetation changes and automatically updating classifications, the CCDC algorithm effectively reduces uncertainties inherent in traditional classification methods and significantly improves temporal consistency, providing key technical support for generating long-term vegetation maps of the QTP. The CCDC is an "online" algorithm that utilizes all available Landsat observations to detect "breaks" in the time series, allowing for the generation of land cover maps for any given time (Zhu and Woodcock, 2014; Zhu et al., 2019). While initially developed for Landsat data, this algorithm has also been used for MODIS and Sentinel datasets (Shimizu et al., 2019;

Tang et al., 2019; Xian et al., 2022). The CCDC algorithm comprises two components: "Continuous Change Detection (CCD)" and "Classification (C)".

The CCD part uses a time series fitting model (Eq. 6) to analyze observational data and employs the root mean square error (RMSE) of the fitting model and the residuals of new observations to detect changes. The model includes harmonic terms and a slope term, where the harmonic terms capture intra-annual seasonal variations, and the slope term estimates inter-annual changes. When the residuals of multiple consecutive observations exceed a specified threshold, the corresponding time period is flagged as a "break." This process is then repeated to identify subsequent "breaks" until all observations have been evaluated. The CCD segments the entire observation sequence into several subsequences, each separated by "breaks." Each subsequence is associated with its own fitting model and specific fitting coefficients.

$$\hat{\rho}(i, t) = c_{0i} + \sum_{n=1}^{N} \left( a_{ni} \cos \frac{2\pi n}{T} t + b_{ni} \sin \frac{2\pi n}{T} t \right) + c_{1i} t \tag{6}$$

where $\hat{\rho}(i, t)$ is the predicted value of the $i^{th}$ band on Julian day t; T is the average number of days in a year; $a_{ni}$ and $b_{ni}$ are the harmonic coefficients of $n^{th}$ order for the $i^{th}$ band i; $c_{0i}$ and $c_{1i}$ are the intercept and slope coefficients, respectively; and N represents the highest order of harmonics, which is set to 3 in this study.

The second part of the CCDC, the "C," uses the RF method to classify each subsequence based on its fitting coefficients (Zhu and Woodcock, 2014). Due to the unique geographical location and climatic conditions of the QTP, terrian and climate factors significantly influence vegetation distribution (Zhou et al., 2023). Therefore, instead of focusing solely on the differences in subsequence coefficients from the CCD results, all detected "breaks" were used to identify potential change areas. Subsequently, this study classified vegetation using an RF model based on terrain-climate-remote sensing data.

The CCDC algorithm was initially developed in MATLAB (Zhu and Woodcock, 2014) and later in Python (Brown et al., 2020). While its "online" advantage enables effective detection of changes and model updates as new observations are collected, this advantage comes with high computational resource demands, particularly for long-term monitoring over extensive areas. The GEE platform, with its robust cloud computing capabilities, resolves these challenges by providing the CCDC algorithm as a time segmentation tool labeled "ee.Algorithms.TemporalSegmentation.Ccdc" (Arévalo et al., 2020; Pasquarella et al., 2022). On the GEE platform, the CCDC algorithm includes parameters such as "breakpointBands," "minObservations," "chiSquareProbability," "minNumOfYearsScaler," "dateFormat," "lambda," and "maxIterations." While some studies have fine-tuned these parameters for better results (Awty-Carroll et al., 2019; Brown et al., 2020; Cohen et al., 2020), this study primarily used the default parameters provided by GEE, modifying only "dateFormat" and "breakpointBands." The "breakpointBands" parameter specifies the bands for breakpoint detection, including R (Red) band, N (Near-infrared) band, and S1(Shortwave Infrared 1) band, which correlate with chlorophyll content, leaf structure, and water content, respectively. The "dateFormat" parameter is set to 2, indicating that all breakpoint times are represented as Unix timestamps.

### 2.4.2 Potential Vegetation Change Area Identification Algorithm

Identifying potential change areas annually is essential for achieving vegetation mapping each year. For example, in dynamically updating the 2019 vegetation map based on the 2020 vegetation map (Fig. 2b), the study area was divided into 4 types (Eq. 7): areas without breakpoints in both 2019 and 2020 (unchanged areas), areas with breakpoints in both 2019 and 2020 (PC 1), areas with breakpoints in 2019 but not in 2020 (PC 2), and areas without breakpoints in 2019 but with breakpoints in 2020 (PC 3). The three types of areas with breakpoints were combined as potential areas for vegetation type changes in

2019, while other areas remained the same as the vegetation types in 2020.

$$S_{T-1} = (B_{T-1} \cap B_T) \cup (B_{T-1} \cap \overline{B_T}) \cup (\overline{B_{T-1}} \cap B_T) \tag{7}$$

where $S_{T-1}$ is the potential vegetation change area for year T-1, $B_{T-1}$ is the breakpoint area for year T-1, $B_T$ is the breakpoint area for year T, $\overline{B_{T-1}}$ is the non-breakpoint area for year T-1, and $\overline{B_T}$ is the non-breakpoint area for year T.

### 2.4.3 Actual Vegetation Type Identification Algorithm

The potential vegetation change areas merely indicate regions where vegetation types might change, thus necessitating further

identification of the actual vegetation change areas. Based on the RF model constructed in 2020, and combining the terrain, climate, remote sensing data, and potential vegetation change areas of 2019, the actual vegetation types in the potential change areas for 2019 can be determined. Subsequently, the 2019 vegetation map was obtained by overlaying and analyzing this data with the 2020 vegetation map (Fig. 2c).

$$R'_{T-1} = M_{2020}(F_{T-1}, S_{T-1}) \tag{8}$$

$$R_{T-1} = \begin{cases} R'_{T-1}, & S_{T-1} \text{ and } R'_{T-1} \neq R_T \\ R_T, & \text{otherwise} \end{cases} \tag{9}$$

where $M_{2020}$ is the RF model for the year 2020, $F_{T-1}$ is the vegetation mapping features for the year T-1, $S_{T-1}$ is the potential

vegetation change area for the year T-1, $R'_{T-1}$ is the vegetation classification result for the potential change area of the year T-1, $R_T$ is the vegetation classification result for the year T, and $R_{T-1}$ is the vegetation classification result for the year T-1.

### 2.4.4 Spatial-temporal Consistency

In this study, RF model is used to generate annual vegetation maps. Although RF handles complex data structures efficiently, its pixel-by-pixel classification method can result in salt-and-pepper noise. To mitigate this issue, this study employs a spatial-

temporal constraint method, which assesses the consistency of each pixel's label within a 3×3×3 cube across both spatial and temporal dimensions (Fig. 2d). The consistency, $C_{x,y,t}$ , is calculated by averaging the agreement of the central pixel's label with the labels of 27 surrounding pixels (Yang and Huang, 2021; Li et al., 2015):

$$C_{x,y,t} = \frac{1}{27} \sum_{i=x-1}^{x+1} \sum_{j=y-1}^{y+1} \sum_{k=t-1}^{t+1} I(\text{Label}_{x,y,t} = \text{Label}_{i,j,k}) \tag{10}$$

where I is an indicator function returning 1 if the labels match, otherwise 0.

If $C_{x,y,t} < 0.5$ and $t > 2000$, the pixel is deemed misclassified, and its label is corrected to match the previous year's central label. For the year 2000, where previous year data is unavailable, the label is adjusted to the most frequent label in the spatial 3×3 area. This approach assumes that significant, inconsistent changes are unlikely both spatially and temporally, thereby enhancing the accuracy of the classification.

## 3 Results

### 3.1 Vegetation Mapping of the QTP at 500 m Resolution in 2020

Due to the significant collinearity among the 126 features in MODIS data, this study focused on 13 features with VIF below 30, including various band reflectance features and vegetation indices such as NDGlaI 15%, NDGlaI 90%, IBI 30%, NDBI 90%, IBI 15%, IBI 90%, EVI 90%, NDVI 90%, IBI 75%, IBI 60%, LSWI 90%, M 90%, and IBI 45%. Combined with two climate factors (AT and AP) and three terrain features (Elevation, Slope, and Aspect), an 18-feature set was created for OOB error analysis. The results indicated that the OOB error decreased as the number of features increased, reaching a minimum of 0.135 with 11 features, beyond which there was a slight increase (Fig. 3a). Subsequently, the top 11 features were selected to construct the RF model: AP, Elevation, NDVI 90%, AT, Slope, M 90%, EVI 90%, NDBI 90%, NDGlaI 90%, NDGlaI 15%, and IBI 75%. Among them, AP, Elevation, AT, and Slope ranked 1st, 2nd, 4th, and 5th in importance score, respectively. Therefore, climate and terrain are significant factors influencing vegetation distribution on the QTP (Fig. 3b). Finally, the 2020 vegetation map of the QTP was generated on the GEE platform at a 500 m spatial resolution (Fig. 4) based on the developed RF model.

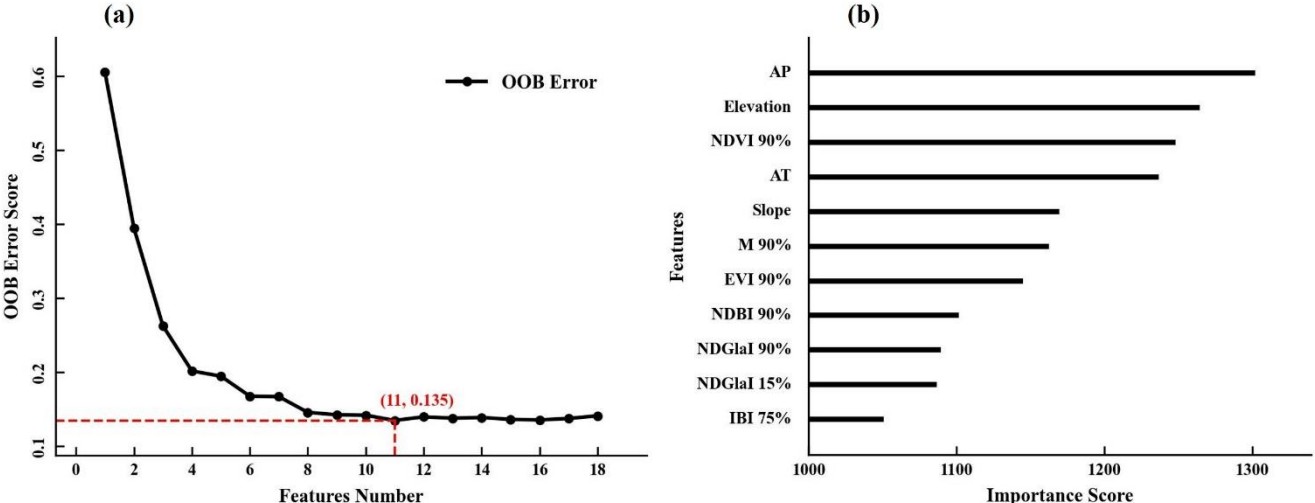

**Figure 3. Evaluation of the 2020 vegetation classification model on the QTP.**

The 2020 vegetation map of the QTP includes 16 types (Fig. 4). Among these, four types cover areas larger than 400,000 km². AM, AG, AD, and AV. The AM covers the largest area, approximately 798,000 km², mainly in the eastern and southern regions of Gansu and Qinghai provinces. The AG covers around 578,000 km², predominantly in the central part of the plateau, with a distinct transitional zone adjoining the AM. The AD, covering about 562,000 km², is primarily located in the Xinjiang region. The AV covers approximately 474,000 km² and is widely distributed from above the treeline or shrub zone to the lower limit of the perennial snow line on the QTP. Among the five forest types in the QTP, ECF cover the largest area, about 154,000 km², mainly distributed in the Hengduan Mountains in the southeast of the QTP and near the Indus River in the west. DCF cover the smallest area, approximately 12,000 km², primarily located along the northwestern border of the QTP.

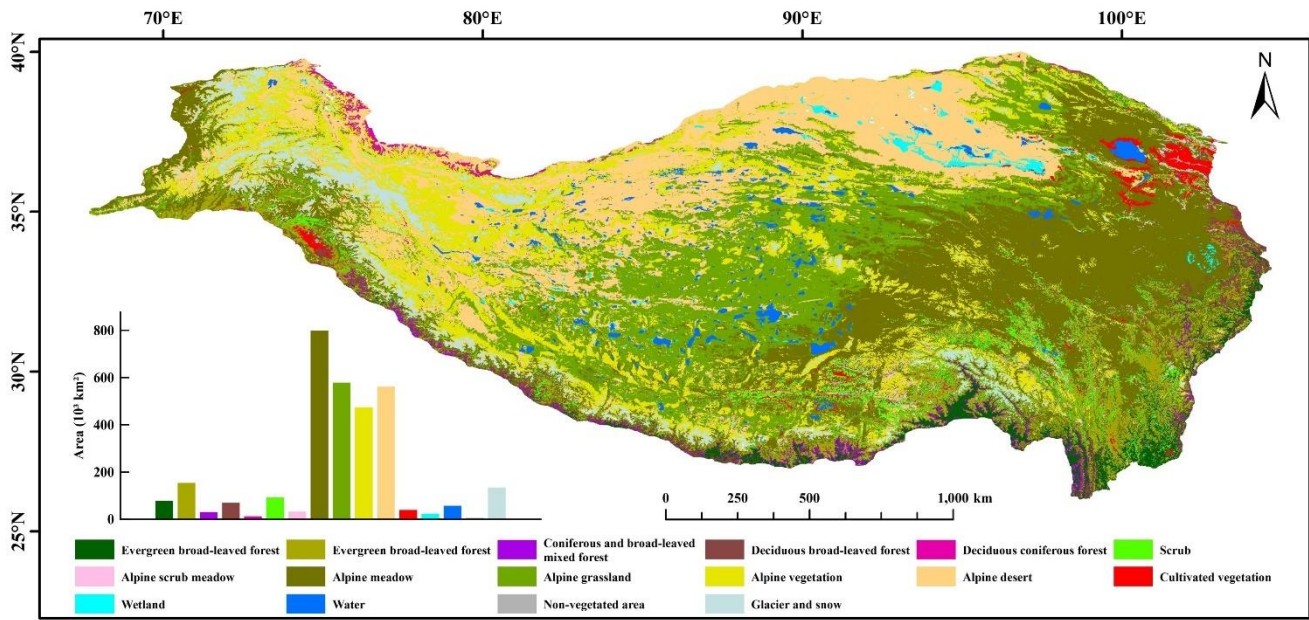

**Figure 4. Vegetation types and spatial distribution at 500 m resolution on the QTP in 2020.**

## 3.2 Continuous Annual Vegetation Mapping at 500 m Resolution from 2000 to 2022

This study used the CCD algorithm to detect "breaks" in the dense time series MODIS data of the QTP from 2000 to 2022, defining these breaks as potential changes (PC) (Fig. 5a). Subsequently, an RF model incorporating terrain, climate, and remote sensing data was applied to annually update and generate vegetation maps of the QTP, identifying changes between adjacent years as real changes (RC) (Fig. 5b).

Approximately 31.8% of the QTP is classified as PC areas. Regions with one or two instances of PC account for 18.2% and 7.12% of the area, respectively (Fig. 5g), primarily located at the junctions of AM, AG, and AD in the central part of the plateau (Fig. 5a). Notably, areas near Hala Lake in the northeast (ROI1) and Mapam Yumco in the southwest Ali region (ROI2) experienced more than five instances of change, covering approximately 0.82% of the total area of the QTP (Fig. 5c-d).

Compared to the PC areas, the RC areas from the annual vegetation maps of the QTP are significantly smaller (Fig. 5b), encompassing only 3.17% of the total area. As shown in Fig. 5g, regions with one or two instances of RC account for 2.75% and 0.34% of the QTP, respectively. Areas with three or more instances of RC make up only 0.08%, with no regions

experiencing five or more instances of change. In all categories of change frequency, the area of RC is consistently smaller than that of PC. In the RC regions of ROI1 and ROI2, only a few areas show one or two instances of change. Notably, in ROI2, no areas with five instances of PC (red) appear in the corresponding RC regions; only a small portion shows a single instance of RC change.

Although the CCD model, which relies on dense spectral data, often detects many false changes—meaning that many

"breaks" detected by CCD do not necessarily represent land type transformations (Zhu et al., 2019)—the RF model based on climate, terrain, and remote sensing data produces annual vegetation maps that effectively reduce these false changes. The frequency and area of detected change regions are significantly smaller compared to the CCD model results.

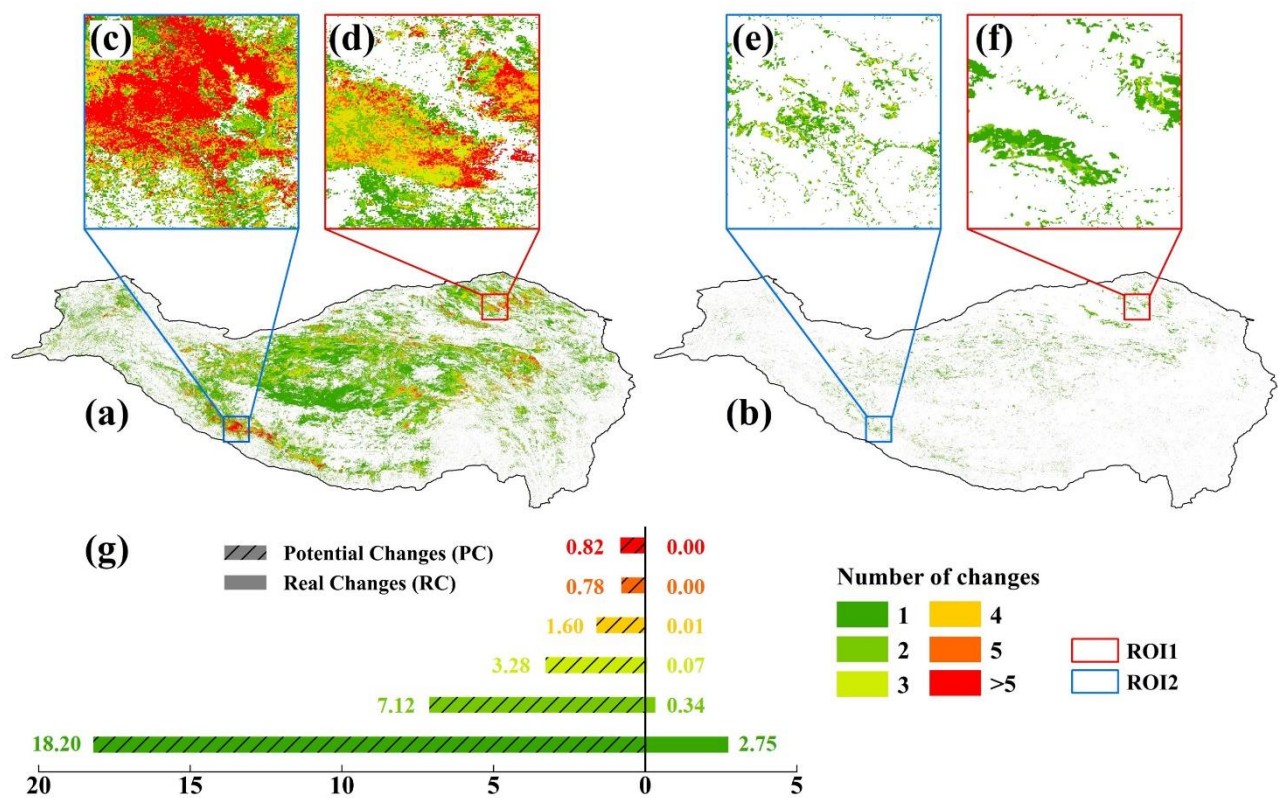

**Figure 5. Overview of potential change (PC) and real change areas (RC) on the QTP from 2000 to 2022. (a-b) Distribution of PC**

**and RC on the QTP; (c-f) PC and RC areas in ROI1 and ROI2; (g) Area statistics of PC and RC on the QTP.**

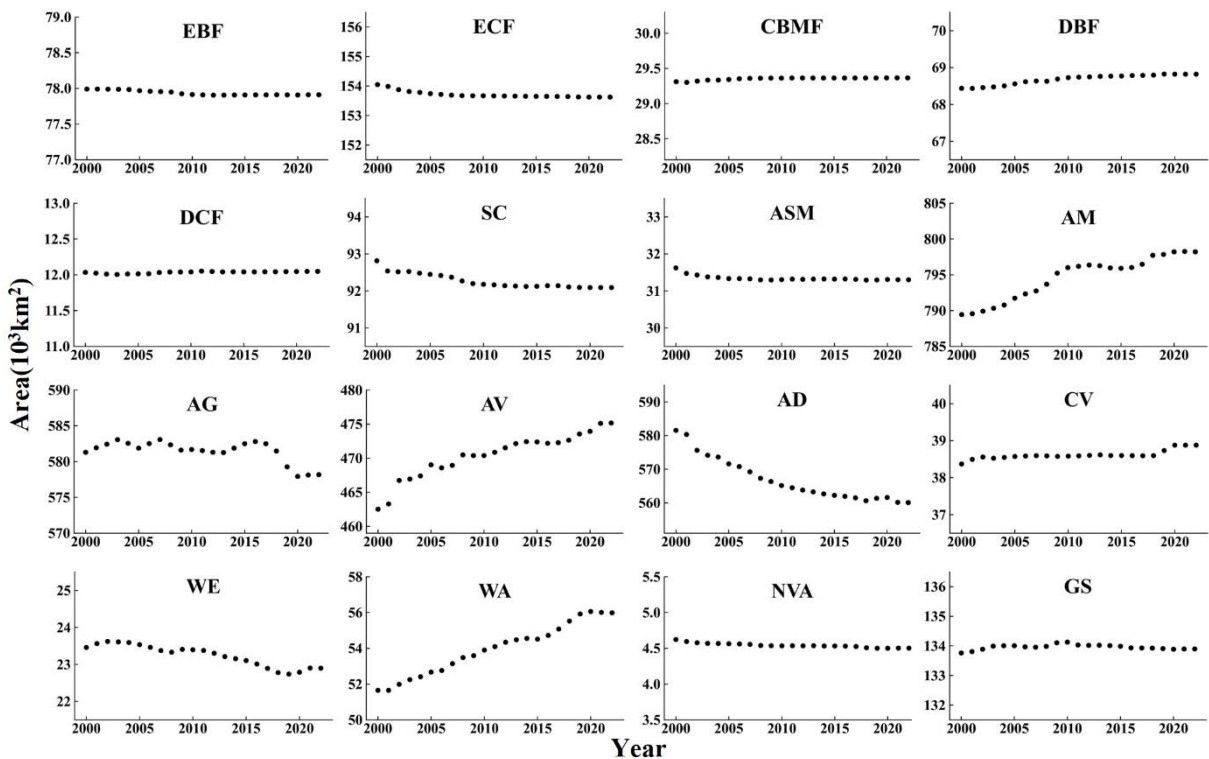

**Figure 6. Annual area changes for 16 vegetation types on the QTP from 2000 to 2022.**

Fig. 6 illustrates the area changes of 16 vegetation types on the QTP from 2000 to 2022. All forest types remained relatively stable, with changes within 0.3%. Among the five forest types, ECF showed a small decrease, reducing by approximately 400 km² (0.28%), while DCF had a small increase by about 14 km² (0.12%). Except for DBF, SC and ASM have limited distributions on the plateau. ASM decreased by about 300 km² (1.02%), primarily before 2005. AM, AG, AV, and AD are the four most widespread types on the plateau. AM and AV showed significant increases, with AM rising by about 8,800 km² (1.11%) and AV by about 12,600 km² (2.73%). In contrast, AD significantly decreased by about 21,500 km² (3.69%). AG fluctuated but notably declined by about 3,100 km² (0.54%) after 2016. CV increased by approximately 500 km² (1.32%), mainly after 2018. WA showed the most significant change, increasing by about 4,300 km² (8.40%) between 2000 and 2022. GS decreased by approximately 200 km² (0.17%), primarily after 2010. Both WE and NVA showed a decreasing trend, with reductions of about 600 km² and 100 km², respectively, from 2000 to 2022.

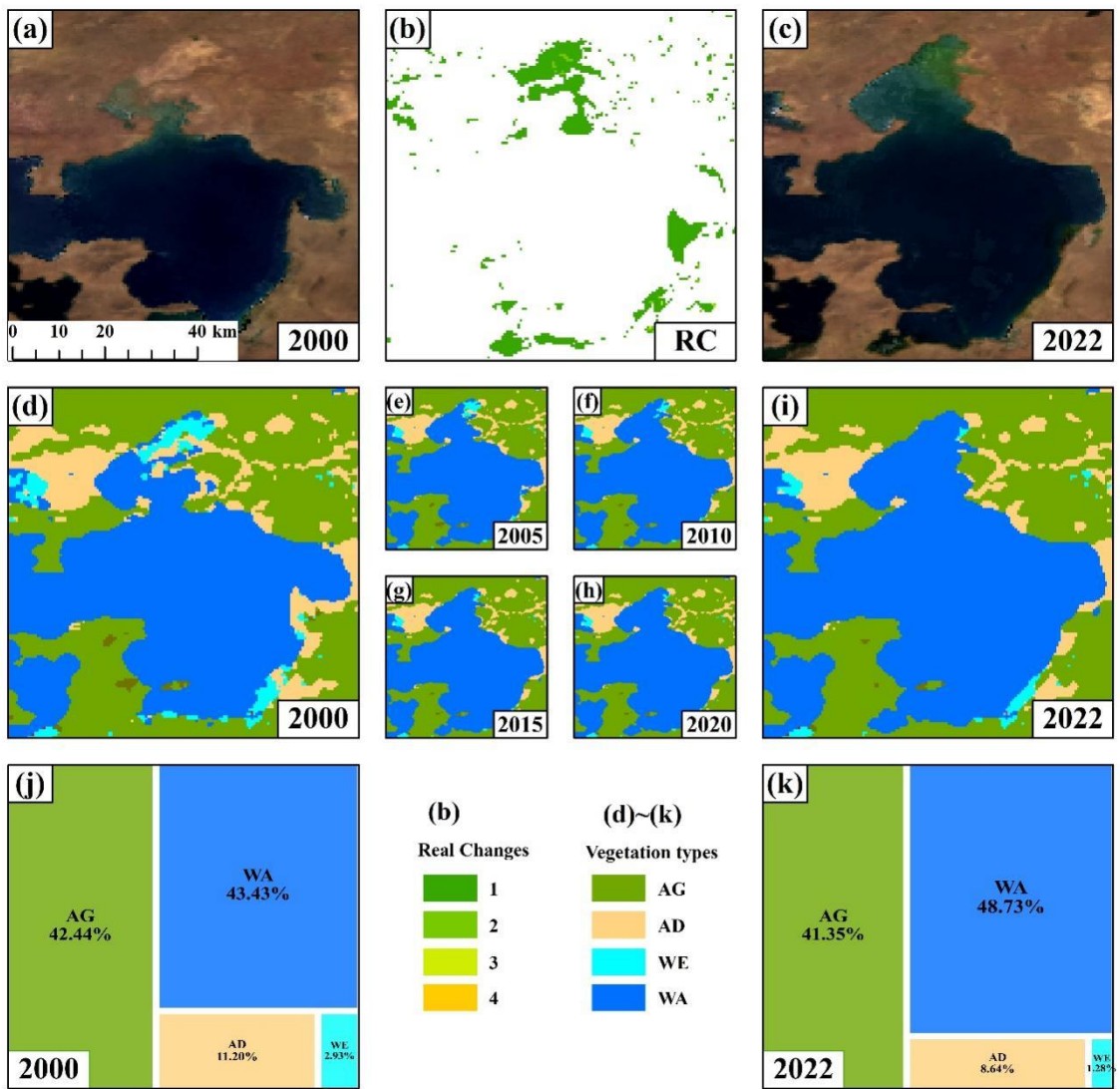

**Figure 7. Annual vegetation cover changes in Area 1 [88.6985E, 31.5002N]. (a, c): Remote sensing images from 2000 and 2022, respectively. (b): Distribution and frequency of RC. (d-i): Annual vegetation maps for 2000, 2005, 2010, 2015, 2020, and 2022. (j, k): Area proportions of major vegetation types in 2000 and 2022, respectively.**

As shown in Fig. 7, most RC occurred only once, predominantly in the northern, eastern, and southwestern parts of Selin Co. In 2000, these areas were classified as WE, AD, and AG, respectively. From 2000 to 2022, these regions gradually transformed into WA, as evident in the remote sensing images. Specifically, the northern WE transitioned to WA between 2000 and 2015, the eastern AD between 2000 and 2010, and the southern AG between 2000 and 2005. The WA in this region increased from approximately 43.43% in 2000 to 48.73% in 2022, marking a 5.3% rise. Simultaneously, AG, AD, and WE

decreased by 1.09%, 2.56%, and 1.65%, respectively. The eastern AD was the most significant contributor to the expansion of Selin Co's water.

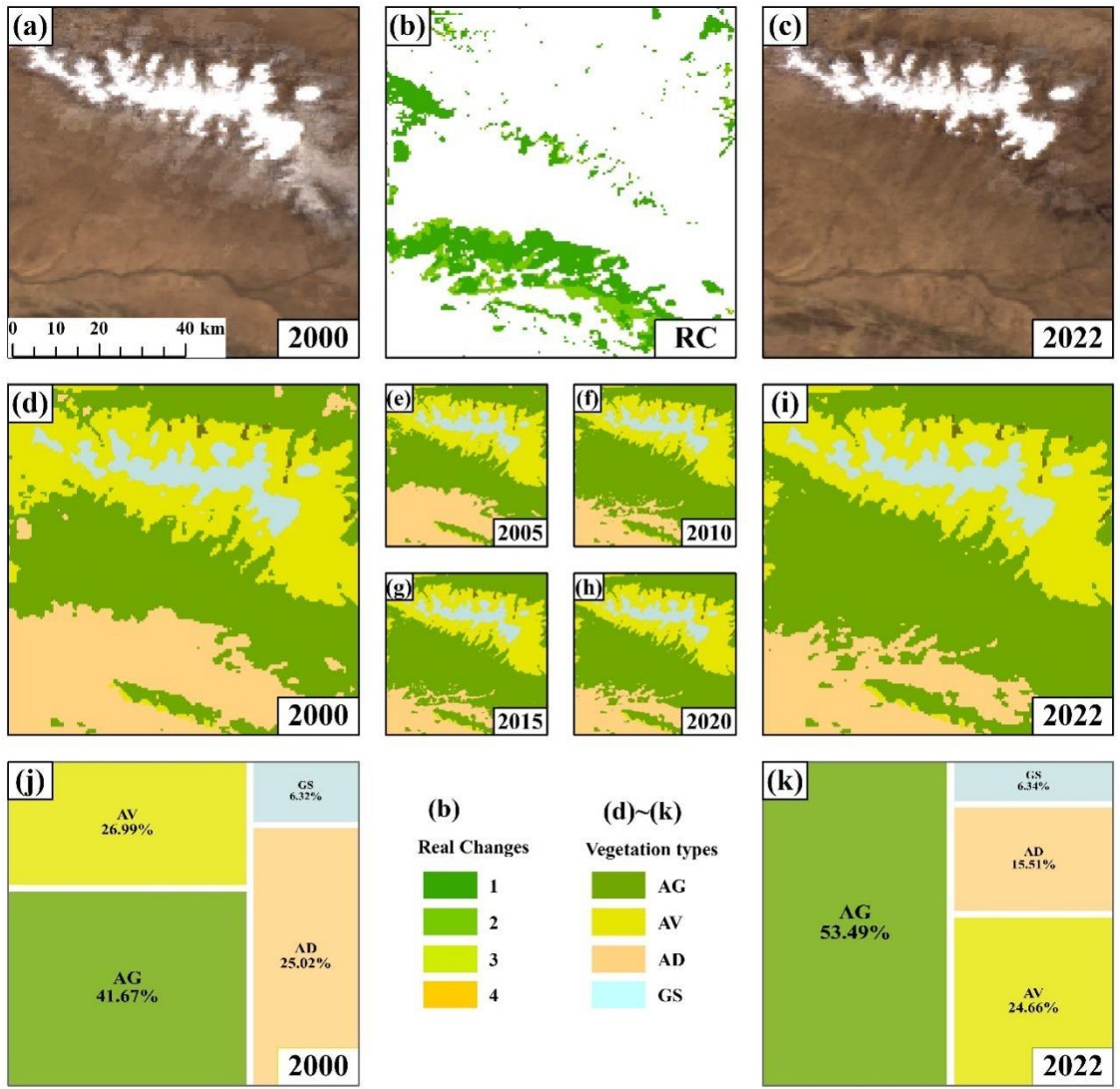

 **Figure 8. Annual vegetation cover changes in Area 2 [95.8644E, 37.6311N]. (a, c): Remote sensing images from 2000 and 2022, respectively. (b): Distribution and frequency of RC. (d-i): Annual vegetation maps for 2000, 2005, 2010, 2015, 2020, and 2022. (j, k): Area proportions of major vegetation types in 2000 and 2022, respectively.**

In the northeastern part of Hala Lake, close to ROI1, the area primarily consists of AG, AV, AF, and GS. Fig. 8b shows that most changes in this region from 2000 to 2022 occurred once or twice. The area of GA changed very little over the past 23 years, increasing slightly from 6.32% to 6.34%. In contrast, AG expanded significantly, from 41.67% to 53.49%, an increase of about 11.82%. Meanwhile, the areas of AD and AV decreased by 9.51% and 2.33%, respectively. These changes mainly

occurred in transition zones, particularly between AG and AD, where most conversion from AD to AG happened between 2000 and 2010. The shift from AV to AG mostly occurred between 2000 and 2005. Notably, some areas in Fig. 8b experienced two changes. These changes mainly reflect fluctuations between AD and AG (Fig. 8c-d), likely due to similar climate and spectral characteristics in the transition zones, causing some pixels to switch between the two types multiple times.

## 3.3 Accuracy Validation

The annual vegetation map of the QTP is dynamically updated using 2020 as the reference year, making its accuracy crucial for long-term product reliability. Due to minimal differences in vegetation distribution between adjacent years, samples from 2019, 2020, and 2021 were used to validate the reference map. This validation set included 1,175 samples. The results showed an OA of 83.32% for 2020, with a Kappa coefficient of 0.82 (Table 4). For most vegetation types, the UA for DCF, AM, AV, AD, and WA exceeded 90%, and the PA for DCF, WE, WA, and GS was also above 90%. However, the UA and PA for CBMF were only 57.14% and 48.00%, respectively. This lower accuracy is likely because CBMF, as a mix of coniferous and broadleaf forests, is often misclassified by the RF model as EBF, ECF, or DBF due to spatial similarities. Overall, the 2020 vegetation map, validated with 1,175 samples, achieved sufficient accuracy to be used as the reference for dynamic updates.

Table 4. Confusion matrix, PA, and UA of vegetation mapping at 500 m resolution on the QTP in 2020

| | EBF | ECF | CBMF | DBF | DCF | SC | ASM | AM | AG | AV | AD | CV | WE | WA | NVA | GS | Total | UA (%) |
|---|---|---|---|---|---|---|---|---|---|---|---|---|---|---|---|---|---|---|
| EBF | 56 | 11 | 8 | 1 | 0 | 0 | 0 | 0 | 0 | 0 | 0 | 0 | 0 | 0 | 0 | 0 | 76 | 73.68 |
| ECF | 2 | 74 | 3 | 2 | 1 | 1 | 0 | 3 | 0 | 0 | 0 | 0 | 0 | 0 | 0 | 0 | 86 | 86.05 |
| CBMF | 0 | 4 | 12 | 4 | 0 | 0 | 0 | 0 | 0 | 0 | 0 | 1 | 0 | 0 | 0 | 0 | 21 | 57.14 |
| DBF | 5 | 4 | 2 | 51 | 0 | 0 | 0 | 0 | 0 | 0 | 0 | 0 | 0 | 0 | 0 | 0 | 62 | 82.26 |
| DCF | 0 | 0 | 0 | 0 | 87 | 0 | 0 | 0 | 2 | 0 | 2 | 0 | 0 | 0 | 0 | 0 | 91 | 95.60 |
| SC | 0 | 0 | 0 | 0 | 0 | 46 | 2 | 6 | 1 | 0 | 1 | 2 | 0 | 0 | 0 | 0 | 58 | 79.31 |
| ASM | 5 | 0 | 0 | 0 | 0 | 8 | 29 | 4 | 1 | 0 | 0 | 1 | 0 | 0 | 0 | 0 | 48 | 60.42 |
| AM | 0 | 0 | 0 | 0 | 0 | 0 | 1 | 54 | 2 | 0 | 0 | 0 | 0 | 0 | 0 | 0 | 57 | 94.74 |
| AG | 0 | 1 | 0 | 0 | 0 | 4 | 0 | 8 | 61 | 0 | 7 | 2 | 0 | 0 | 0 | 0 | 83 | 73.49 |
| AV | 0 | 0 | 0 | 0 | 0 | 0 | 1 | 2 | 5 | 89 | 0 | 0 | 0 | 0 | 0 | 0 | 97 | 91.75 |
| AD | 0 | 0 | 0 | 0 | 5 | 1 | 0 | 1 | 1 | 0 | 85 | 0 | 0 | 0 | 0 | 0 | 93 | 91.40 |
| CV | 0 | 1 | 0 | 1 | 0 | 1 | 0 | 6 | 0 | 0 | 2 | 87 | 0 | 0 | 7 | 0 | 105 | 82.86 |
| WE | 0 | 0 | 0 | 0 | 2 | 0 | 0 | 6 | 1 | 0 | 4 | 3 | 60 | 0 | 0 | 0 | 76 | 78.95 |
| WA | 0 | 0 | 0 | 0 | 0 | 1 | 0 | 3 | 0 | 0 | 0 | 0 | 0 | 83 | 0 | 0 | 87 | 95.40 |
| NVA | 0 | 2 | 0 | 0 | 0 | 1 | 0 | 2 | 0 | 0 | 0 | 9 | 0 | 0 | 22 | 0 | 36 | 61.11 |
| GS | 0 | 0 | 0 | 0 | 0 | 0 | 0 | 0 | 0 | 16 | 0 | 0 | 0 | 0 | 0 | 83 | 99 | 83.84 |
| Total | 68 | 97 | 25 | 59 | 95 | 63 | 33 | 95 | 74 | 105 | 101 | 105 | 60 | 83 | 29 | 83 | 1175 | |
| PA (%) | 82.35 | 76.29 | 48.00 | 86.44 | 91.58 | 73.02 | 87.88 | 56.84 | 82.43 | 84.76 | 84.16 | 82.86 | 100.00 | 100.00 | 75.86 | 100.00 | | 83.32 |

The third-party samples (1,014 samples) and visual interpretation samples (1,002 samples) were used to validate the accuracy of annual vegetation maps. The third-party samples mainly include grasslands, AD, and GS. Because the First All-season Sample Set and the Global Land Cover Validation Samples did not differentiate between AM and AG, these two types were combined under the grasslands category during the validation process. The third-party samples mainly originate from

2014 (190 samples) and 2015 (666 samples), with smaller numbers from other years. The visual interpretation samples exhibit stable characteristics across multiple years, making them suitable for validating annual vegetation maps from 2000 to 2019.

Validation results indicated that accuracy from 2000 to 2019 remained above 80%, with the lowest being 82.07% in 2015 (Fig. 9). Upon inspection, most of the misclassified samples were located in the Barren land (vegetation cover <10%) category in the First All-season Sample Set, which corresponds to AD in the vegetation maps. However, some samples of this category were classified as AG in the vegetation map. There are definitional differences between Barren land and AD. Additionally, AD and AG represent two largest vegetation types on the QTP, and there are significant transitional zones between these two types (Fig. 4). As a result, the accuracy in 2015, when a large number of third-party samples were used, was slightly lower than in other years. Moreover, because there were insufficient validation samples for 2021 and 2022, and 2020 has already been validated as the reference year, the overall accuracy of the annual vegetation maps is estimated to be 83.27%, based on the average accuracy from 2000 to 2019.

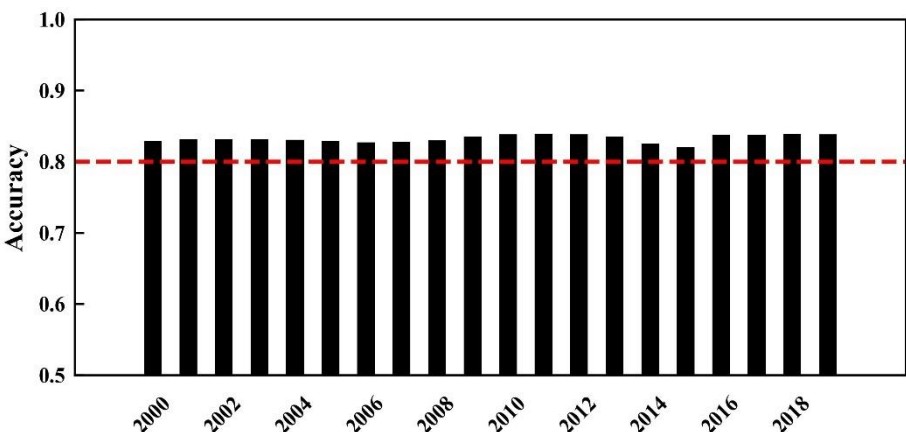

Figure 9. Annual overall accuracy of vegetation maps from 2000 to 2019.

## 4 Discussion

### 4.1 Evaluating the Efficacy of the CCD Algorithm in Annual Vegetation Mapping from 2000 to 2022

This study proposed a method for long-time continuous annual vegetation mapping. Specifically, the CCD algorithm was applied to MODIS data from 2000 to 2022 to detect breakpoints. Subsequent processes involved identifying potential change areas, recognizing true vegetation types, and spatial-temporal consistency. This enabled consistent mapping of vegetation on the QTP annually from 2000 to 2022. The CCD algorithm in this study used harmonic functions to fit long-term remote sensing images, thereby identifying breakpoints and determining the timing of these breakpoints. To ensure the algorithm effectively captures vegetation dynamics, we selected the R (Red), N (Near-infrared), and S1(Shortwave Infrared 1) bands for breakpoint detection. The R band is related to chlorophyll content, the N band to canopy structure, and the S1 band to vegetation water

content (Tucker, 1979; Curran, 1989; Gao, 1996). Combining these complementary bands enables the CCD algorithm to effectively detect breakpoints in vegetation dynamics on the QTP.

For instance, in Fig. 10, the CCD algorithm was applied to detect changes in the R, N, and S1 bands of the sampling site from 2000 to 2022. The results indicated that there was a breakpoint in 2011 (highlighted in yellow), dividing the period into Fit1 (from 2000 to 2011) and Fit2 (from 2011 to 2022). The annual amplitude of the three bands exhibited small fluctuations with stable interannual patterns. In contrast, the amplitude of the R and N bands in Fit2 far exceeded that of Fit1, showing significant differences in seasonal patterns within the year. Additionally, for the S1 band, the overall reflectance in Fit2 was

substantially lower than in Fit1, reflecting distinct seasonal fluctuations within the year. Based on the annual Landsat images from 2009 to 2013, there was a noticeable expansion of WA in the selected area, marking a transition from AD prior to 2011 to WA thereafter. The annual vegetation maps accurately captured this transition, with the area represented as brown-yellow (AD) before and up to 2010, and shifting to deep blue (WA) from 2011 onward. This change was further corroborated by the CCD fitting results and visual interpretation of the Landsat images.

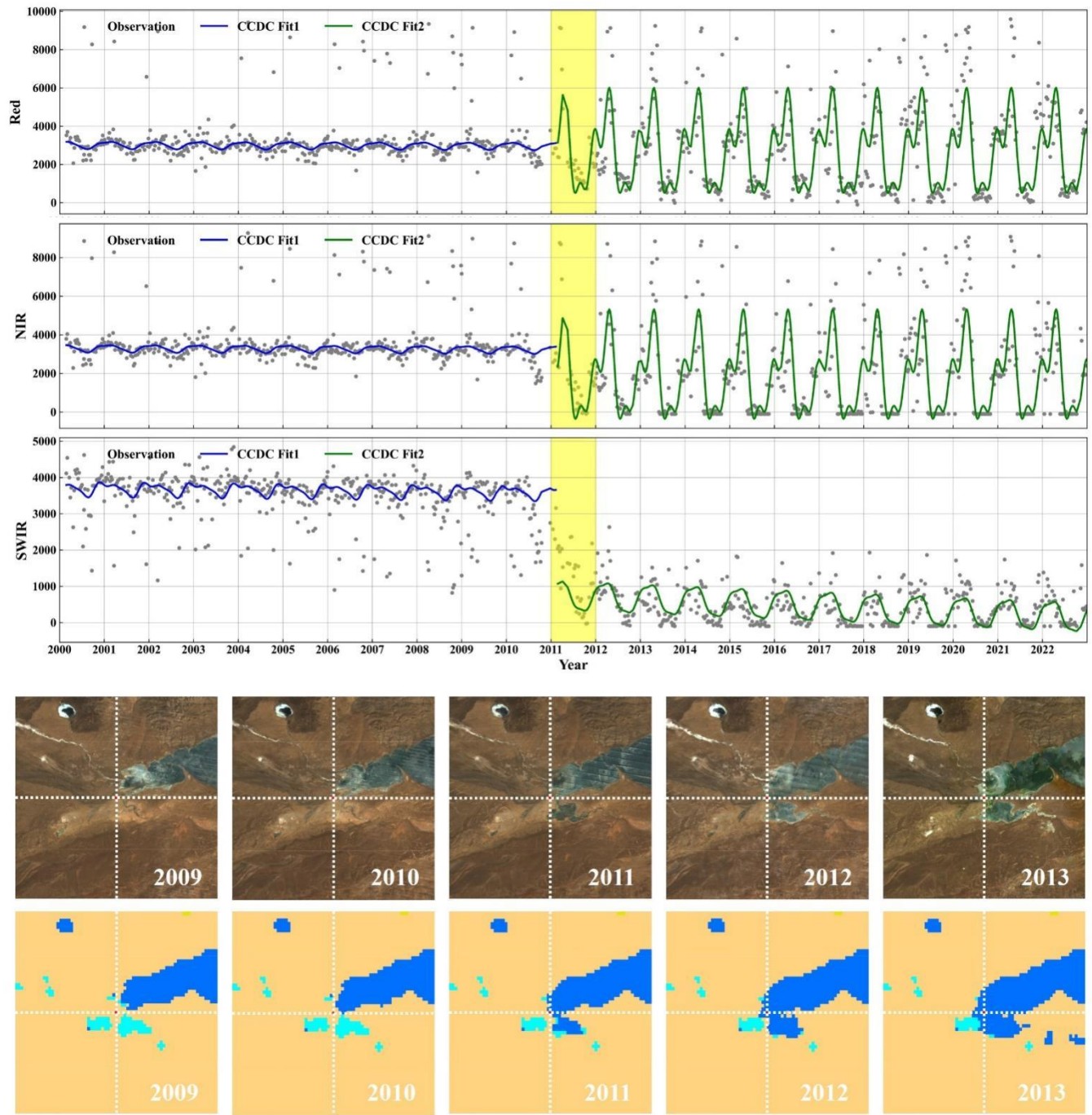

**Figure 10. Validation of vegetation mapping: consistent change detection in WA sample [82.9294E, 35.2425N] by CCD, Landsat, and annual vegetation maps.**

In Fig. 11, the CCD algorithm indicated that there were no breakpoints from 2000 to 2022. The annual and interannual variations in the R, N, and S1 bands were stable over the years. Combined with the Landsat images, the selected area was

445 consistently classified as EBF without any detected changes. The long-time annual vegetation maps consistently reflected this characteristic.

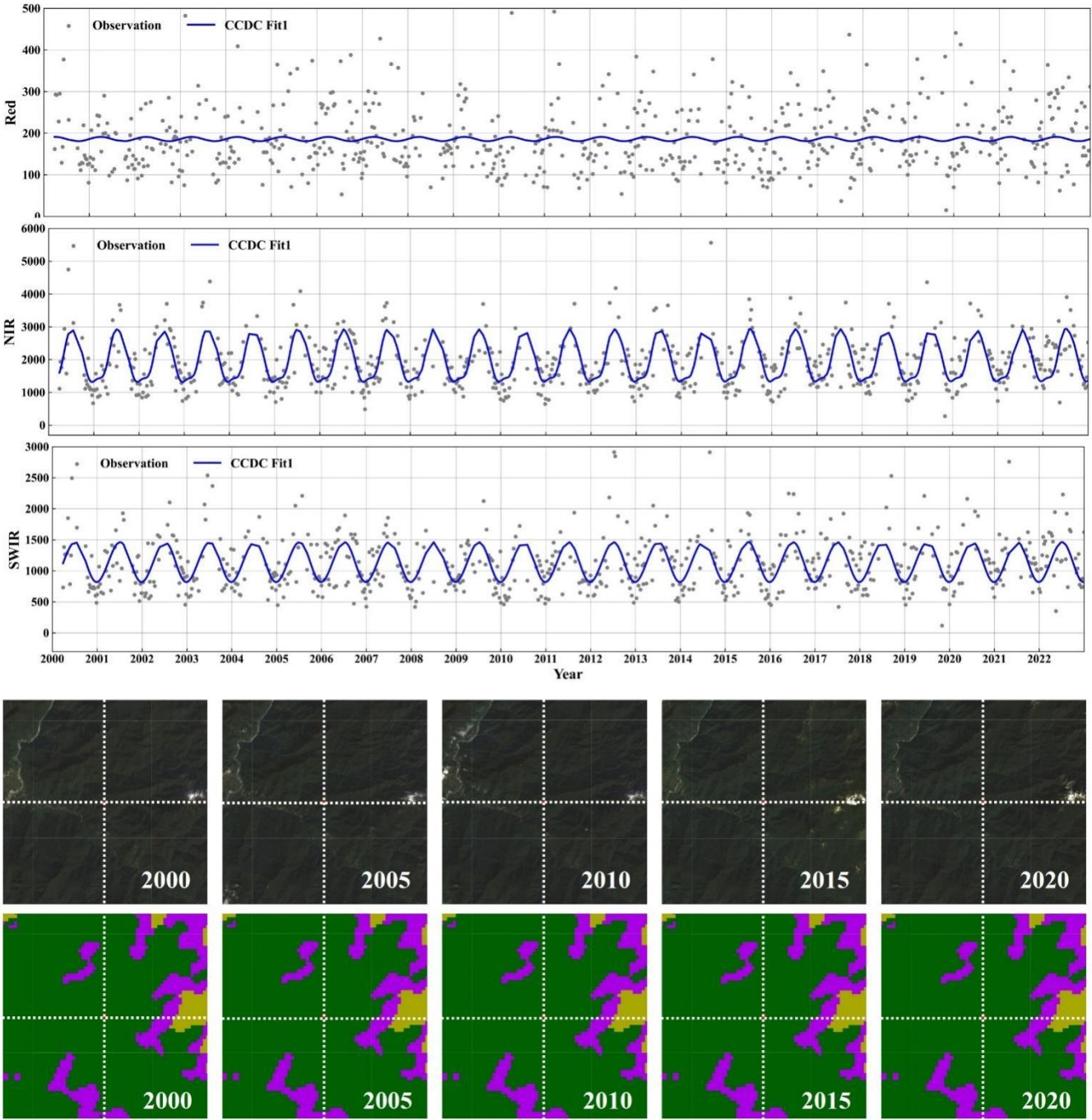

**Figure 11. Validation of vegetation mapping: consistency of no change detection in forest sample [95.2794E, 28.7617N] by CCD, Landsat, and annual vegetation maps.**

However, abrupt changes detected by the CCD algorithm may not always accurately reflect real changes on the ground (Zhu and Woodcock, 2014; Du et al., 2023). In Fig. 12, an area of CV on the QTP was examined. The CCD results indicated a breakpoint in 2018, with data from 2000 to 2018 categorized as Fit1 and post-2018 as Fit2. Although Fit2 shows greater amplitude in the R, N, and S1 bands compared to Fit1, the waveform remains similar. Despite a breakpoint was detected, Landsat images from 2016 to 2020 confirm that the area consistently featured CV. The detected changes were likely caused
by variations in cultivation practices in 2018 or similar factors, rather than actual changes in vegetation. As a result, the annual vegetation maps did not capture the change detected by the CCD in 2018.

Given its performance, the CCD algorithm effectively identifies the regions and timings of breakpoints in long-time remote sensing imagery, making it a suitable foundational method for detecting potential change areas in annual vegetation mapping. Although the CCD algorithm is susceptible to false positives due to factors such as changes in cultivated species (Fig. 12), the
subsequent methods employed in this study, including potential area identification, true vegetation type recognition, and spatial-temporal constraints, help mitigate these false positive errors (Fig. 12).

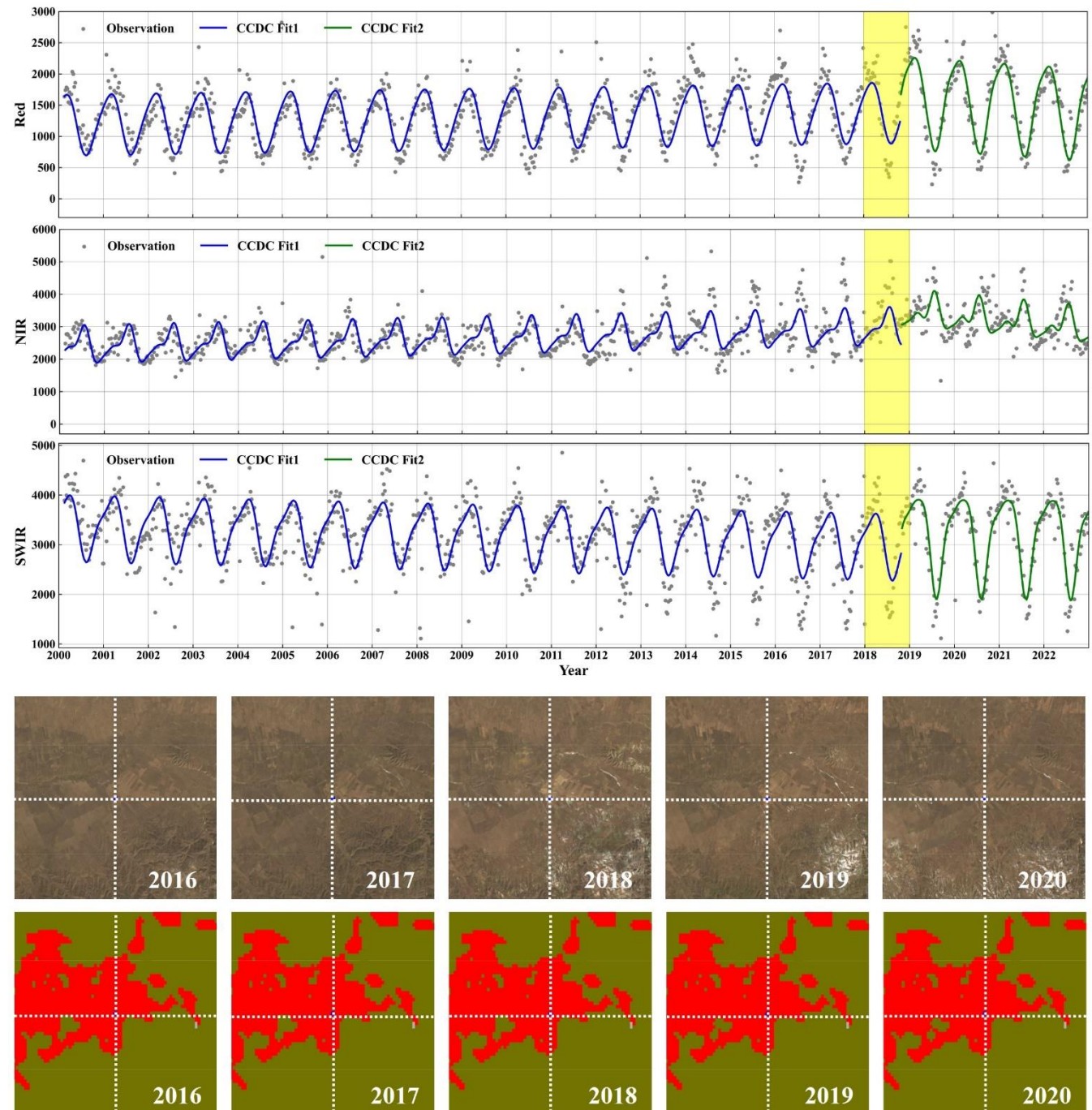

**Figure 12. Validation of vegetation mapping: CV sample [100.8570E, 35.2474N] changes detected by CCD not reflected in Landsat and annual vegetation maps.**

## 4.2 Inter-comparison with other products

MCD12Q1 (Sulla-Menashe et al., 2019), CLCD (Yang and Huang, 2021), and GLC-FCS30D (Zhang et al., 2024b) were selected for cross-comparison with the annual vegetation maps. MCD12Q1 provides annual global land cover data at a 500 m resolution from 2001 to 2022, using the International Geosphere-Biosphere Programme (IGBP) classification system, which includes 17 categories such as evergreen needleleaf forest, grassland, and cropland. CLCD offers 30 m land cover data for China from 1990 to 2022, classifying land into nine types: cropland, forest, shrub, grassland, water, snow/ice, barren, impervious, and wetland. GLC-FCS30D delivers land cover data from 1985 to 2022, with five-year intervals before 2000 and annual intervals thereafter, covering 35 detailed land cover categories. Based on the temporal coverage of these products, annual vegetation maps from 2001, 2011, and 2021 were selected for cross-validation, with a regional focus on four main types: WA, GS, grassland, and forest.

Landsat remote sensing images reveal significant shrinkage of water in this region from 2001 to 2021 (Fig. 13a). In 2001, all four products accurately depicted the distribution of WA, with clear boundaries and complete representation. By 2011, GLC-FCS30D identified part of the WA as WE, while the other three products consistently showed it as WA. By 2021, the WA had markedly shrunk compared to their extent in 2001. Compared to the Landsat images, CLCD showed an omission of WA, with most areas transforming into AD, while GLC-FCS30D misclassified part of the WA as WE and identified some non-water areas as WA.

For GS areas, all four products demonstrated good identification in 2001, 2011, and 2021 (Fig. 13b). Uniquely, the annual vegetation maps identified additional "alpine vegetation". Consequently, in the vicinity of GS, MCD12Q1 categorized these areas as AD, while CLCD and GLC-FCS30D largely classified them as grassland. The annual vegetation maps not only preserved the distribution details of GS, AD, and AG from the comparison products but also provided a more detailed classification of AV.

However, the three comparison products classified this region simply as grassland without further differentiation (Fig. 14a). Grassland is a dominant vegetation type on the QTP, covering approximately 57% of the QTP. According to Zhou et al. (2022a), grasslands on the QTP can be divided into AG, AM, ASM, and AD. Therefore, it is essential to differentiate between AG, AM, ASM, and AD. In Fig. 14a, the overall distribution of AG and AM has remained consistent over time, though changes have occurred in their transition zones. For example, in the southwestern AM region of Fig. 14a, some AG areas have gradually transitioned to AM over the past 23 years.

Forests are sparsely distributed on the QTP, found mainly on the southern slopes of the Himalayas and in the southeastern region (Fig. 14b). Their distribution has remained stable over the years with minimal change. Among the four products, CLCD did not differentiate forest types, and thus represents them under the general category of 'Forest'. In contrast, the other three products classified the forested areas as EBF. MCD12Q1 shows a smaller forest extent and includes a notable proportion of CBMF within this range. The forest extent in the annual vegetation maps, CLCD, and GLC-FCS30D shows high consistency with the Landsat images. However, the annual vegetation maps identified some ASM in the meadow areas adjacent to forests.

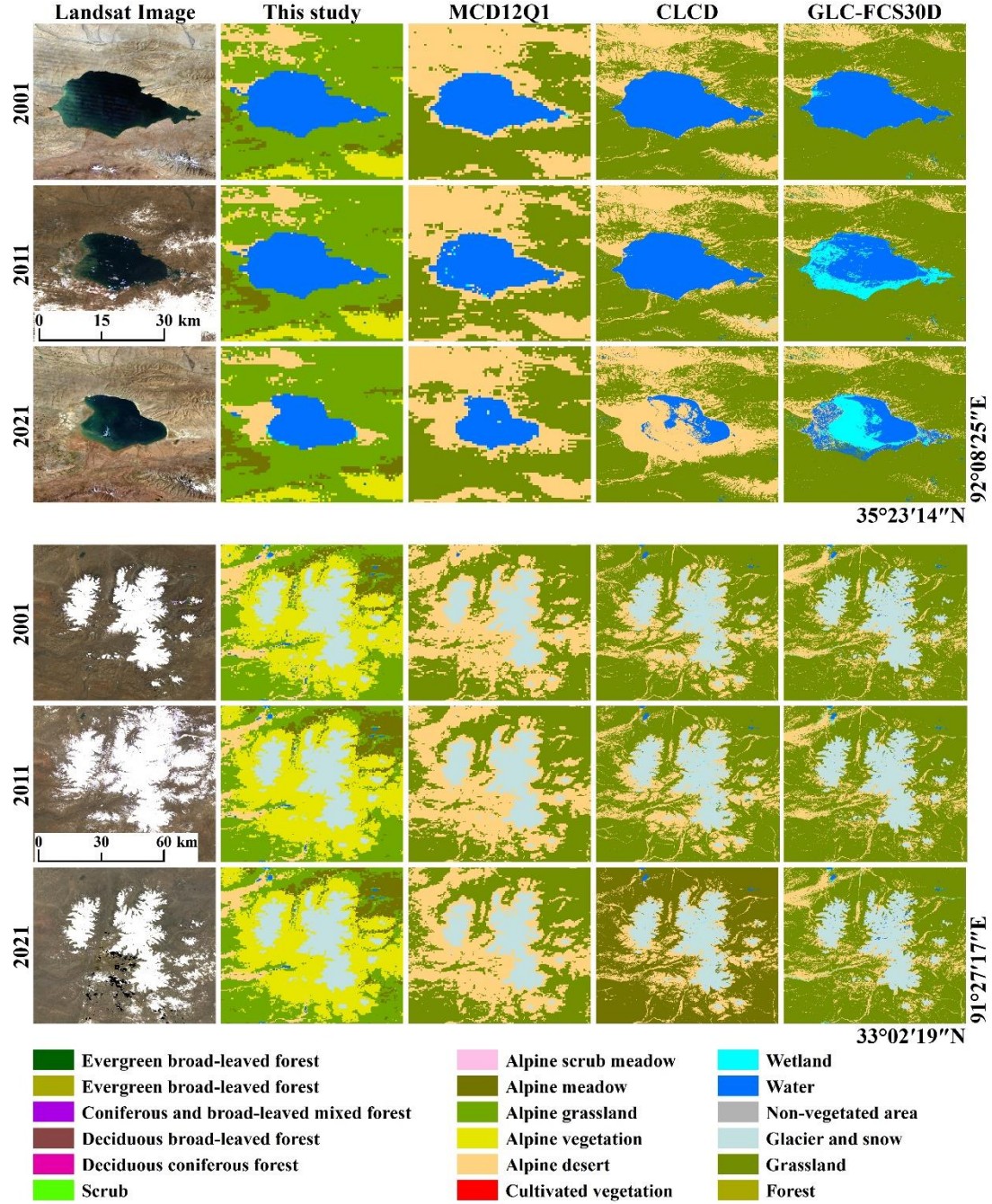

**Figure 13. Comparison with the annual vegetation maps, MCD12Q1, CLCD, and GLC-FCS30D. (a): Region dominated by WA and AG. (b): Region dominated by GA and AM.**

500

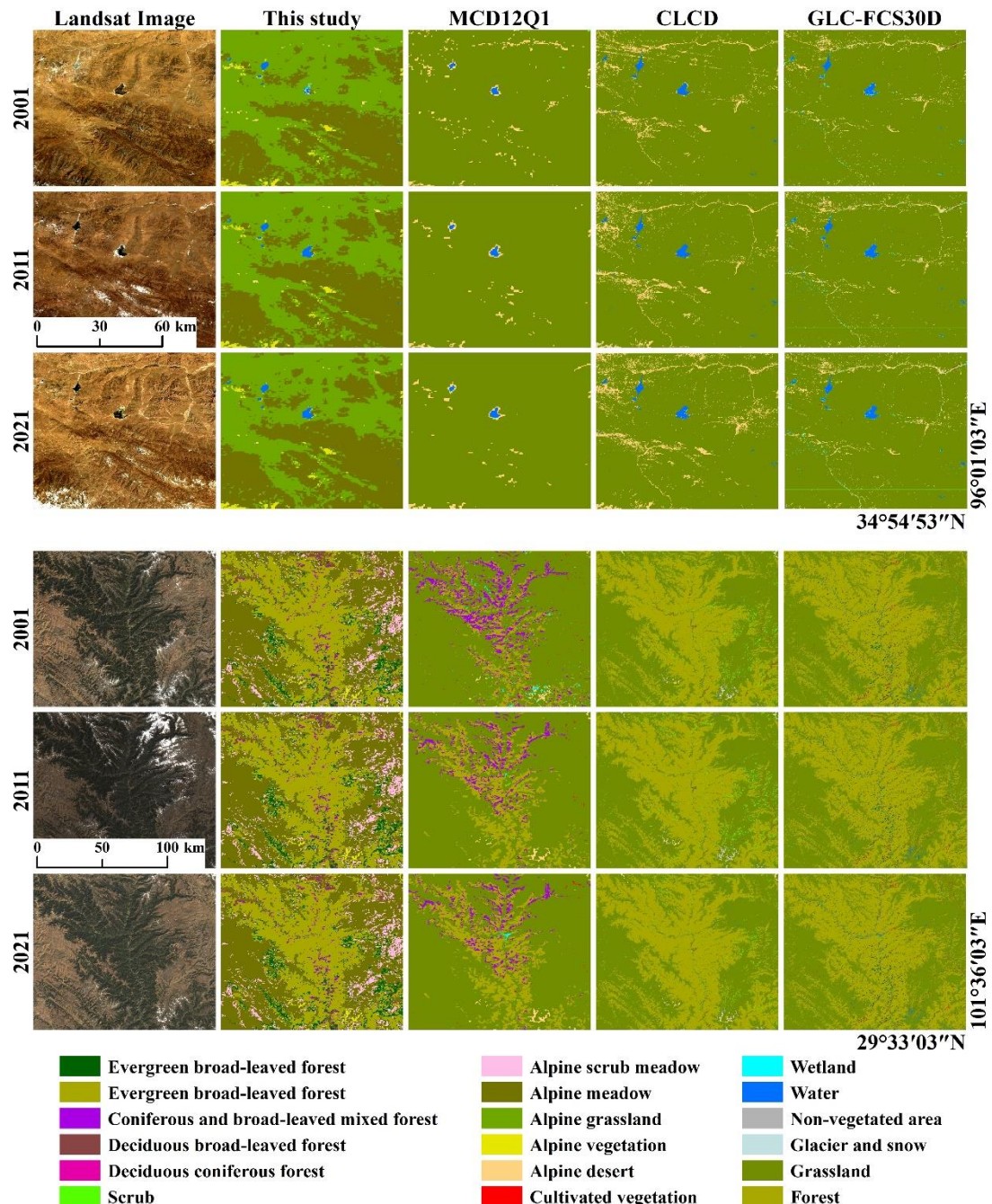

**Figure 14. Comparison with the annual vegetation maps, MCD12Q1, CLCD, and GLC-FCS30D. (a): Region dominated by AM and AG. (b): Region dominated by EBF and AM)**

## 5 Data availability

The 500 m annual vegetation maps of QTP from 2000 to 2022 are available at https://data.tpdc.ac.cn/en/disallow/6304c1a4-efc0-4766-bae3-4148bdf7bcfd (Zhou et al., 2024). The vegetation maps are stored in TIFF format, with the file name "QTP_Vegetation_Map_XXXX.tif," where XXXX represents the year. All files can be opened and reprocessed using software such as ArcGIS, QGIS, and ENVI. Each TIFF dataset contains values from 0 to 16, where 0 represents invalid values, and 1 to 16 correspond to the 16 vegetation types listed in Table 1. The MOD09A1, MCD12Q1, and SRTM data used in this study were obtained from GEE (http://code.earthengine.google.com, last access: 23 July 2024). The "1-km monthly precipitation dataset for China (1901-2023)", the "1-km monthly mean temperature dataset for China (1901-2023)", the "Vegetation map of Qinghai Tibet Plateau in 2020 with 10 m spatial resolution", and QTP boundary dataset were provided by the National Tibetan Plateau / Third Pole Environment Data Center (https://data.tpdc.ac.cn, last access: 29 September 2024). The First All-season Sample Set can be download from https://data-starcloud.pcl.ac.cn/resource/54 (last access: 29 September 2024). The Global Land Cover Validation Samples can be download from https://zenodo.org/records/3551995 (last access: 29 September 2024). The CLCD can be download from https://zenodo.org/records/8176941 (last access: 23 July 2024), and the GLC-FCS30D can be download from https://zenodo.org/records/8239305 (last access: 23 July 2024).

## 6 Conclusions

Long-time series of annual regional vegetation types and geographic distribution data are vital for examining the impact of climate change on vegetation and its evolutionary trends. In this study, annual vegetation of the QTP from 2000 to 2022 at a 500 m spatial resolution was mapped through the MOD09A1 product, together with a reference year vegetation classification map and a breakpoint detection algorithm. The study achieved an overall accuracy of 83.27% for continuous annual vegetation mapping at a 500 m resolution from 2000 to 2022. The study supports the use of remote sensing data to mapping a long-term continuous annual vegetation. Furthermore, it facilitates the elucidation of the spatial and temporal evolution of regional and global vegetation under the background of global warming.

## Author contributions

GZ and HR conceived the study. LZ and HR analysed the data and drafted the paper. XL and MZ supported the generation of the dataset and the analysis of the results.

## Competing interests

The authors declare that they have no conflict of interest.

**Financial support**

This work was supported by the Second Tibetan Plateau Scientific Expedition and Research Program (grant number
2019QZKK0106) and the National Natural Science Foundation of China (grant number 42141007).

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
