# Peer review of "Annual vegetation maps in Qinghai-Tibet Plateau (QTP) from 2000 to 2022 based on MODIS series satellite imagery"

_Earth System Science Data, 2024_

## Author Response (AR1)

**Responses to Reviewers Comments**

**Dear Reviewer #1:**

This study developed a new approach to long-time continuous annual vegetation mapping from remote sensing imagery. The new approach is very important and effective for the use of remote sensing data to mapping long-term continuous annual vegetation. Using the new method, this study mapped the vegetation of Qinghai Tibet Plateau(QTP) from 2000 to 2022 at a 500 m spatial resolution through the MOD09A1 product. The valuable dataset facilitates the understanding of the spatial and temporal variations of vegetation in the QTP under the background of global warming. Therefore, I recommend accepting this manuscript. However, there are some minor issues that need to be addressed:

**Q1.**L39-40 Modify "Over time, the trend of increasing misclassified pixels and their positional uncertainties ultimately reduces the reliability of remote sensing interpretation for LCSV types." to "Over time, there is a trend of increasing misclassified pixels and their positional uncertainties, which ultimately reduces the reliability of remote sensing interpretation for LCSV types."

   **Answer:** The sentence is modified based on the comment in the revised manuscript. Please see Lines 40-42 in the revised manuscript.

**Q2.**L104-106 Sentence structure is incomplete. The suggestion is as follows: "The MOD09A1 dataset provides surface reflectance in seven spectral bands (Red, Blue, Green, NIR, MIR, SWIR 1, and SWIR 2) with 500 m spatial resolution, and all cloud-contaminated pixels are removed."

   **Answer:** The sentence structure is modified based on the suggestion in the revised manuscript. Please see Lines 122-123 in the revised manuscript.

**Q3.**L114 Use "were" instead of "was" to match the plural noun "data": "For the QTP within China, climate data at 1,000 m were obtained from..."

   **Answer:** It is modified in the revised manuscript in the Line 131. Also, "climate data for areas of the QTP outside China was derived...." was modified by using "were" instead of "was" in the Line 132.

**Q4.**L301 The verb "confirms" should be corrected to "confirm" to match the plural subject.

**Answer:** The verb "confirm" is used in the revised manuscript. Please see Line 383.

*Q5.*(5) In the Figure 4, legend text is too small, please revise the font size.

**Answer:** In the revised manuscript, the Figure 4 is deleted based on the comments from Reviewer #2.

**Thanks very much for these constructive comments and suggestions.**

**Dear Reviewer #2:**

This manuscript combined the continuous change detection classification (CCDC), random forest (RF) classification models, and "Spatial-temporal Consistency" algorithm to develop the annual vegetation maps in QTP during 2000-2022 using time-series MODIS imagery. The results seem good with an overall accuracy of 86.5% and a kappa of 0.85. However, there are some serious concerns in the current manuscript:

*Q1:* The proposed method lacks of the novelty, the "CCDC+RF+ Spatial-temporal Consistency" has been applied in some previous works, such as: Xian et al. (2022) and Friedl et al. (2022). All of them used this strategy to generate the time-series global/national land-cover change maps at 30 m from Landsat imagery.

**Answer:** Thank you very much for the constructive comments and suggestions. While the method proposed in this study is based on "CCDC+RF+ Spatial-temporal Consistency", it differs from the approaches used by Xian et al. (2022) and Friedl et al. (2022). Firstly, the CCDC algorithm comprises two parts: "Continuous Change Detection (CCD)" and "Classification (C)"(Awty-Carroll et al., 2019; Zhu and Woodcock, 2014). Previous works often used the full CCDC, utilizing coefficients (spectral-temporal features) of each sub-series marked by CCD for classification through an RF model to determine vegetation types. However, our results showed that climate and terrain factors significantly influence vegetation distribution on the QTP. Therefore, we use only the "Break" points marked by the CCD to identify potential change (PC) areas on the QTP. The actual category labels are determined by the RF model, which is based on climate, terrain, and remote sensing data, rather than the sub-series model coefficients used in previous studies.

At present, the land-cover change maps in the QTP from other scientists primarily targets land use types and lacks detailed vegetation categorization. Our study designed a vegetation classification system tailored to the unique vegetation types of the QTP. For instance, we have refined the classification for the Tibetan Plateau by dividing grassland into three distinct categories: alpine meadow, alpine grassland, and alpine desert. Additionally, we provided a detailed classification of forest into five categories: evergreen broad-leaved forest, evergreen coniferous forest, coniferous and broad-leaved mixed forest, deciduous broad-leaved forest, and deciduous coniferous forest.

*Q2:* The descriptions about the CCDC algorithm should be strengthen. The CCDC can take advantage of continuous satellite observations and achieve higher robustness for capturing

land-cover changes, however, it was a parameter-sensitive model, please refer the works of Zhu et al. (2019). So you should emphasize how to determine these parameters in your manuscript.

**Answer:** Thanks very much for the constructive comments and suggestions. We have strengthened the description of the CCDC in the revised manuscript; please refer to pages 9-10, lines 183-214. The CCDC is a parameter-sensitive model. Zhu et al. (2019) provided a set of well-performing parameters for the CCDC, which have been adopted as default parameters in GEE. These include "chiSquareProbability=0.99", "minObservations=6", "lambda=20", and several others. We modified the "breakpointBands" and "dateFormat", while the others remain at default settings. The "breakpointBands" parameter specifies the bands used for breakpoint detection, including Red, NIR, and SWIR 1, which correlate with chlorophyll content, leaf structure, and water content, respectively. The "dateFormat" parameter is set to 2, indicating that all breakpoint times are represented as Unix timestamps.

*Q3:* The applicability of the CCDC algorithm in the study area is questionable. The work of Pasquarella et al. (2022) clearly stated that the CCDC was suitable for the stationary time-series and is actually designed for forest disturbances. The Figure 6a in this manuscript obviously showed that the CCDC model suffered the under-fitting problems.

**Answer:** Thank you very much for the constructive comments and suggestions. Previous studies have shown that the CCDC algorithm was originally designed to detect land cover and land use changes that usually result in a large change magnitude. However, its accuracy is relatively lower when identifying forest disturbances with subtle spectral changes, particularly when the forest cover type remains unchanged after the disturbance (Cohen et al., 2017; Pasquarella et al; 2022; Zhu et al., 2019). Although Fig. 6a in the original manuscript (Fig. 11a in the revised manuscript) shows under-fitting issues with the CCDC, this problem did not occur in the NIR and SWIR bands. Furthermore, this study aims to identify transitions between different vegetation types, which typically involve more significant changes. Minor forest disturbances, which are challenging for the CCDC to detect, remain classified as forest types before and after the disturbance and thus do not affect our study results. Finally, the under-fitting of the CCD mainly manifests as the inability of the fitting coefficients (spectral-temporal features) to accurately represent the corresponding vegetation types. However, this study only uses the "break" (potential changes) detected by the CCD, and these potential changes significantly decrease after classification using our RF model, which is based on climate, terrain, and remote sensing data.

*Q4:* The validation section also cannot convince me. The validation points came from their 10m vegetation maps only using the purity index, I think the classification errors in the 10m product

would transfer to the validation points. Then, since the training and validation samples came from the same sample pool, the final validation accuracy is also high, which is unconvincing.

**Answer:** Thank you for pointing out the issues in the validation section. Validation samples from the same pool as the training samples can result in overly optimistic validation results. To address this, we removed all the validation samples used in the origin manuscript. Instead, we gathered new validation samples through field surveys, literature reviews, and visual interpretation. For details, please refer to page 4, lines 97-110 in the revised manuscript. For the reference year (2020), we collected 1,382 samples, and for other years, we gathered between 1,100 and 1,200 samples. The validation results show an accuracy of 83.43% for the 2020 vegetation map, with an average accuracy of 81.82% for the annual vegetation maps from 2000 to 2022.

*Q5:* The descriptions about section 3.2 should be greatly revised. The Figure 4 is meaningless because there is no way to know where changes have occurred throughout the study area. I strongly suggest the authors can refer the works of Xian et al. (2022) to reorganize this content, and pay more attention to the analysis of land-cover changed areas.

**Answer:** Thank you for your insightful suggestion. Section 3.2, which covers the annual vegetation mapping results, was minimally detailed in the original manuscript and lacked details on vegetation changes. We reorganized this section by referring to the works of Xian et al. (2022). The revised content is on pages 13-18, lines 271-328. We first analyze vegetation changes across the entire QTP, then examine two sample areas in detail, using remote sensing images and vegetation maps to show vegetation distribution changes from 2000 to 2022.

*Q6:* At present, there are time-series 30 m global land-cover change products, thus, what are the strengths of this data, which need to be emphasized. I also suggest that the manuscript can add the comparisons between their results and some previous datasets such as: CLCD.

**Answer:** Thank you for your suggestion. We have reorganized Section 4.2 to include comparisons with MCD12Q1, CLCD, and GLC-FCS30D, highlighting the strengths of our dataset. We selected four regions with distinct dominant land cover types: water, glaciers and snow, grasslands, and forests. Please refer to pages 24-26 and lines 395-430 in the revised manuscript. Grasslands cover about 57% of the QTP, including AM, AG, ASM, and AD (Zhou et al., 2023). Previous products, such as MCD12Q1, CLCD, and GLC-FCS30D, do not differentiate between these types. Our dataset, however, accurately distinguishes between alpine meadows and alpine grasslands and identifies the unique alpine vegetation, which accounts for about 15% of the QTP.

**References**

Awty-Carroll, K., Bunting, P., Hardy, A., and Bell, G.: An evaluation and comparison of four dense time series change detection methods using simulated data, Remote Sens., 11, 2779, https://doi.org/10.3390/rs11232779, 2019.

Cohen, W., Healey, S., Yang, Z., Stehman, S., Brewer, C., Brooks, E., Gorelick, N., Huang, C., Hughes, M., and Kennedy, R.: How similar are forest disturbance maps derived from different Landsat time series algorithms?, Forests, 8, 98, https://doi.org/10.3390/f8040098, 2017

Friedl, M. A., Woodcock, C. E., Olofsson, P., Zhu, Z., Loveland, T., Stanimirova, R., Arevalo, P., Bullock, E., Hu, K.-T., Zhang, Y., Turlej, K., Tarrio, K., McAvoy, K., Gorelick, N., Wang, J. A., Barber, C. P., and Souza, C.: Medium Spatial Resolution Mapping of Global Land Cover and Land Cover Change Across Multiple Decades From Landsat, Front. Remote Sens., 3, 894571, https://doi.org/10.3389/frsen.2022.894571, 2022.

Pasquarella, V. J., Arévalo, P., Bratley, K. H., Bullock, E. L., Gorelick, N., Yang, Z., and Kennedy, R. E.: Demystifying LandTrendr and CCDC temporal segmentation, Int. J. Appl. Earth Obs., 110, 102806, https://doi.org/10.1016/j.jag.2022.102806, 2022.

Xian, G. Z., Smith, K., Wellington, D., Horton, J., Zhou, Q., Li, C., Auch, R., Brown, J. F., Zhu, Z., and Reker, R. R.: Implementation of the CCDC algorithm to produce the LCMAP Collection 1.0 annual land surface change product, Earth Syst. Sci. Data, 14, 143–162, https://doi.org/10.5194/essd-14-143-2022, 2022.

Zhu, Z. and Woodcock, C. E.: Continuous change detection and classification of land cover using all available Landsat data, Remote Sens. Environ., 144, 152-171, https://doi.org/10.1016/j.rse.2014.01.011, 2014.

Zhu, Z., Zhang, J., Yang, Z., Aljaddani, A. H., Cohen, W. B., Qiu, S., and Zhou, C.: Continuous monitoring of land disturbance based on Landsat time series, Remote Sens. Environ., 238, 111116, https://doi.org/10.1016/j.rse.2019.03.009, 2019.

**Thank the reviewer very much for these constructive comments and suggestions.**

---

## Author Response (AR2)

**Responses to Reviewers Comments**

**Editor**

*Q:* Given the significant concerns raised by the reviewer regarding the originality, it is essential that you thoroughly address all their feedback and make the required revisions to your manuscript. Once you have finalized the revisions, kindly resubmit your manuscript for further evaluation.

Specifically the reviewer has the following comment: "During my review, I found a similar paper on vegetation type mapping in the region, published by the same author team in Scientific Data (https://www.nature.com/articles/s41597-024-03649-7). There appears to be some overlap in the methods and data between the two studies. I am unclear on the rationale for generating annual 500-m vegetation maps in this paper, given that 30-m time series data already exist for the region." Please address this concern.

*Answer:* Liu et al. (2021) pointed out that although existing land cover products meet performance requirements, they fall short in ensuring interannual stability. This is because these products are typically generated independently for each year, which limits their utility for studying land cover changes over time. To address this issue, our team has developed two approaches aimed at improving the stability of annual land cover products.

First, we use visually interpreted and stable samples, which are consistent across multiple years, to generate long-term vegetation maps. This method employs an independent annual generation process but ensures consistency by focusing on stable samples. We applied this approach in our study published in *Scientific Data* (https://www.nature.com/articles/s41597-024-03649-7), where we produced 30 m resolution vegetation maps at ten-year intervals (Ren et al., 2024). While this method uses stable samples to establish classification models, it is still limited by the classifier's inherent misclassification errors, which tend to accumulate spatially over time. In ten-year interval maps, these errors are minimal, but when applied to annual products, the errors compound over 23 years, leading to potentially unpredictable errors.

To overcome this limitation, we propose a second method: a dynamic vegetation mapping approach based on the Continuous Change Detection (CCD) model. First, a reference vegetation map of the QTP for 2020 was produced using terrain-climate-remote sensing data. The CCD model was then applied to detect potential change areas, identifying the temporal and spatial positions of breakpoints. The RF model was used only for pixel-based classification in these breakpoint regions, while non-breakpoint areas were not reclassified. Finally, a spatio-temporal consistency approach was implemented to reduce misclassified pixels, enhancing interannual stability. The key innovation of this method lies in dynamically updating the map each year, rather than classifying the entire

study area annually. By reducing the number of pixels requiring classification, we minimize the risk of misclassification that could accumulate over the 23-year period.

However, the Qinghai Tibet Plateau (QTP) presents unique challenges, particularly in the southeastern forested areas, which are frequently cloud-covered. Landsat observations in these regions are sparse, with fewer than 100 usable images over the 23-year period (R2 Fig. 1a, R2 Fig. 1b). This scarcity is insufficient for CCD, which relies on dense, high-quality time series data. In contrast, MODIS data offer a higher temporal resolution with daily global coverage. Its MOD09A1 product provides 8-day composites, offering a much higher observation frequency compared to Landsat's 16-day revisit cycle (R2 Fig. 1b-d). Despite CCD being originally developed for Landsat, we opted for MODIS data due to its higher temporal coverage, which better suited the mapping objectives for the QTP. Although MODIS's spatial resolution (500 m) is lower, it was the most practical solution for producing consistent annual vegetation maps in this region.

[Figure]

**R2 Figure 1: Frequency of Cloud-Free Pixels from Landsat (a, b) and MOD09A1 (c, d) over the QTP (2000-2022)**

**References**

Liu, L., Zhang, X., Gao, Y., Chen, X., Shuai, X., and Mi, J.: FinerResolution Mapping of Global Land Cover: Recent Developments, Consistency Analysis, and Prospects, J. Remote Sens., 2021, 5289697, https://doi.org/10.34133/2021/5289697, 2021.

Ren, H., Wu, F., and Zhou, G.: The 30 m vegetation maps of the Tibetan Plateau (1990-2020), TPDC [dataset], https://doi.org/10.11888/Terre.tpdc.301126, 2024.

**Thank you very much for the valuable feedback and suggestions.**

**Reviewer #1**

This study presents a novel approach for continuous annual vegetation mapping on the Qinghai-Tibet Plateau (QTP), and achieved good validation accuracy. Detailed vegetation classification data is crucial for ecological management and geographical modeling. According to the previous reviewers' comments, the manuscript has been significantly improved. Here are some issues that need to be addressed.

*Q1:* Line 37 What are the "independent classification methods"? The introduction is too brief, especially lacking a description of other representative methods, which makes it difficult to highlight the advantages of the method used in this study.

   *Answer:* Thanks. We have expanded the explanation of the "independent classification method" by detailing its use in products like GlobCover and MOD12Q1. These examples illustrate the limitations of independently generated land cover maps, particularly in terms of interannual stability. This additional context highlights the advantages of our method in addressing these limitations. The revised content is included in lines 39-53 of the revised manuscript.

*Q2:* Line 8 There is an extra quotation mark in "Third Pole" of the Earth".

   *Answer:* Thank you for pointing this out. We have removed the extra quotation mark from "Third Pole of the Earth" in line 8 of the revised manuscript.

*Q3:* Line 42 Is it "LCSV" or "LUSV"? Please check through the text.

   *Answer:* Thank you. We have reviewed the text and corrected it to "LCSV" (land cover and surface vegetation) in lines 67 and 69 of the revised manuscript.

*Q4:* Several abbreviations are missing their full forms the first time they appear, such as "CLCD".

   *Answer:* Thank you. We have added the full forms of abbreviations like CLCD, GLC_FCS30, GLC_FCS30D, and NASA upon their first mention in lines 30-36 of the revised manuscript.

*Q5:* Line 85 Why was the "Vegetation map of Qinghai Tibet Plateau in 2020 with 10 m spatial resolution" chosen as the source of training samples? There is a lack of description regarding its production method and product accuracy.

   *Answer:* Thank you for your suggestion. The "Vegetation map of Qinghai Tibet Plateau in 2020 with 10 m spatial resolution" was selected because of its overall accuracy of 89.5%, achieved using a regional vegetation mapping method that integrates terrain, climate, and remote sensing data. Its classification system is consistent with the one used in this study, with the exception of glacial snow and ice. We have added a detailed explanation for selecting this product, along with descriptions of

its production method and accuracy, in lines 96-99 of the revised manuscript.

***Q6:*** Line 91 How was the 70% threshold determined?

***Answer:*** Thank you for your question. In Table 1, we provide the area proportions for different purity classes after resampling. Areas with purity levels above 70% cover approximately 62.34% of the QTP (within China). A higher threshold would reduce the available sampling area, making it challenging to sample discrete and smaller vegetation types like shrubs. Conversely, a lower threshold would increase the presence of mixed pixels, affecting the quality of training samples. After careful consideration, we determined that 70% strikes the best balance.

***Q7:*** CCD can detect significant changes, such as the conversion from grassland to water bodies. However, how should more subtle changes between similar vegetation types, such as the transition between alpine meadows and alpine steppes, be managed?

***Answer:*** Thank you for your question. The CCD method can indeed detect significant changes, such as the transition from grassland to water bodies, as shown in Fig. 7 and Fig. 13. In addition, the CCD method can detect transitions between similar vegetation types, such as the shift from alpine meadows to alpine grasslands, by distinguishing subtle differences in their spectral-temporal characteristics. Fig. 14 demonstrates the gradual transition from grassland to meadow in the sample area from 2001 to 2021, primarily occurring in transitional zones between vegetation types. Finally, for areas flagged by CCD as having potential changes, we used a Random Forest (RF) model incorporating terrain, climate, and remote sensing data to verify the changes and classify the vegetation types before and after the transitions.

***Q8:*** Line 44, The sentence is missing a preposition.

***Answer:*** Thanks. We have added the preposition "for" after "critical," and the revised sentence now reads: "The QTP, known as the 'Roof of the World' and the 'Water Tower of Asia', is critical for global climate regulation and regional socio-economic development." This change is reflected in line 54 of the revised manuscript.

**Thank the reviewer very much for these constructive comments and suggestions.**

**Reviewer #2**

This study produced annual 500-m vegetation maps for the Qinghai-Tibet Plateau from 2000 to 2020 by integrating MODIS data, random forest and CCDC algorithms. While this dataset has the potential to support climate regulation and ecosystem conservation efforts in the region, I have significant concerns regarding the data quality, validation process, and methodological innovation after reviewing the manuscript.

**Q1:** ESSD is a journal that emphasizes the publication of original, high-quality datasets. It is unclear why vegetation types were mapped at a relatively coarse 500-m resolution and why the MOD09A1 product was chosen over higher-resolution data like Landsat, particularly considering CCDC was originally developed for Landsat time series.

    **Answer:** The objective of this study is to develop annual vegetation maps for the Qinghai-Tibet Plateau (QTP). However, certain areas, such as the southeastern forests, are frequently cloud-covered. In these regions, Landsat provides fewer than 100 usable images over the 23-year period (R2 Fig. 1a, R2 Fig. 1b), which is insufficient for the CCD algorithm's requirement for dense time series data. In contrast, MODIS, with its daily global coverage and 8-day composite MOD09A1 product, offers a much higher observation frequency compared to Landsat's 16-day revisit cycle (R2 Fig. 1b-d). Although CCD was initially designed for Landsat data, MODIS's higher temporal resolution made it the more suitable choice for the mapping objectives of this study. Despite its lower spatial resolution (500 m), MODIS was the most practical solution for consistently generating annual vegetation maps of the QTP from 2000 to 2022.

[Figure]

**R2 Figure 1: Frequency of Cloud-Free Pixels from Landsat (a, b) and MOD09A1 (c, d) over the QTP (2000-2022)**

***Q2:*** High-resolution vegetation maps have already been produced at 10-m levels (Zhou et al., 2022), and these were used as a baseline to generate samples in this study. This means the samples were extracted from vegetation maps resampled from 10 m to 500 m. For a 500 m × 500 m pixel, significant mixing of different vegetation types is likely. The authors performed random sampling in areas where purity exceeded 70%, but this raises the question: why exclude mixed areas? What were the classification results in these mixed areas?

***Answer:*** Thank you for your valuable suggestion. The presence of mixed pixels at the 500-meter scale is indeed a common issue in vegetation classification. In such cases, the vegetation type of a mixed pixel is typically determined by the dominant vegetation type at the sub-pixel level. To account for this, we set a threshold of 70% purity to select samples, ensuring that the dominant vegetation type occupies a significant portion of the pixel.

In addition, we conducted a correlation test to examine the impact of pixel purity on classification accuracy. Pixels with a purity greater than 70% were defined as high-purity, while those below 70% were considered low-purity. The results showed that models built from high-purity samples provide reliable guidance for classifying low-purity pixels. Specifically, when low-purity pixels were used as validation samples, the model's classifications aligned with the dominant vegetation type of those pixels. However, models built from low-purity samples tended to misclassify high-purity pixels. resulting in lower accuracy compared to models built from high-purity samples and validated with high-purity pixels.

In Table 1, we present the distribution of different purity classes and their area proportions after resampling. Approximately 62.34% of the QTP (within China) consists of areas with purity levels above 70%. We chose 70% as a threshold because setting it too high would reduce the area available for sampling, especially for discrete and smaller vegetation types like shrubs. Conversely, a lower threshold would increase the proportion of mixed pixels, which would negatively affect the quality of training samples.

In summary, the vegetation type of mixed pixels depends on the dominant vegetation type at the sub-pixel scale, and models built from high-purity samples can provide guidance for classifying low-purity areas, while models built from low-purity samples tend to mislead the classification of high-purity areas.

***Q3:*** Regarding data validation, I find the process insufficiently rigorous, which is a crucial element for a data-focused paper. The samples, spanning 2000–2020, were derived from the literature, but the description of this process is vague. How were the samples defined and extracted? How was

their quality assessed? What was their spatial coverage?

*Answer:* Thank you for your suggestion. The quality of validation samples is indeed crucial for the accuracy assessment of our product. Based on your suggestion, we have updated the validation sample dataset and included a more detailed description of the quality control measures implemented.

We collected third-party samples from several sources, including the First All-season Sample Set produced by Li et al. (2017), the Global Land Cover Validation Samples collected by Zhang et al. (2021), and additional validation samples gathered through literature review. To ensure the quality of these samples, we applied the quality control method outlined by Feng et al. (2012).

Specifically, at the spatial scale, we analyzed the homogeneity of validation samples using a 3×3 grid of nine 500 m resolution pixels. The maximum and minimum values for several spectral bands (blue, green, red, near-infrared, shortwave infrared 1, and shortwave infrared 2) were used to calculate the difference, which served as an indicator of homogeneity. The thresholds for homogeneity were 0.03 for the blue, green, red, and shortwave infrared bands, and 0.06 for the near-infrared band. If the range for all bands within a 3×3 grid fell below these thresholds, the area was considered homogeneous. This ensures that at most one representative sample is present within a 500 m spatial extent for the third-party validation samples, and that this sample lies within a 1.5 km×1.5 km homogeneous region.

After implementing these quality control measures, we finalized a set of 1,014 third-party validation samples, consisting of 327 from the Global All-season Sample Library, 499 from the Global Validation Sample Set, and 188 from literature search.

*Q4:* Additionally, these samples were used to validate static maps, not the change maps. In the Discussion, the authors claim that CCDC detects higher frequencies of vegetation type changes (e.g., 5 times), but they also argue that the maps generated in this study are more accurate due to fewer detected changes. How can this claim be substantiated when the accuracy of the change maps was not assessed?

*Answer:* Thank you for your suggestion. The validation samples used in this study can only assess the accuracy of static vegetation maps, and therefore cannot be applied to evaluate change maps. The term "more accurate" in the text refers specifically to a reduction in false positive errors compared to the potential change areas detected by the CCD algorithm. While the CCD algorithm may detect a breakpoint, it does not always correspond to an actual change in vegetation type.

To support this conclusion, we provide an example from Fig. 12. The CCD algorithm detected a breakpoint in this area; however, based on our mapping method and visual interpretation of remote sensing imagery, the area remains classified as cultivated vegetation. As a result, the vegetation

changes detected in this study exhibit a smaller spatial extent and fewer occurrences than the potential changes identified by the CCD method (Fig. 5). This suggests that our maps are more accurate when compared to the CCD detection results, as they avoid false detections of vegetation change.

*Q5:* Moreover, methodological innovation is a key issue for publication in ESSD. As another reviewer noted, the "RF+CCDC+Spatial-temporal Consistency" framework is a common mapping approach in the existing literature, which limits the method novelty of this study.

*Answer:* Thank you for your comment. As Liu et al. (2021) noted, while existing land cover products generally meet user requirements in terms of performance, they often fail to ensure interannual stability because they are generated independently for each year. The "RF+CCDC+Spatial-temporal Consistency" framework is an effective approach to address this issue. The CCD model segments long time series remote sensing data at 'breakpoints', creating sub-sequences with different coefficients. In the study by Xian et al. (2021), CCD output parameters and auxiliary data were used as variables in a classification model, which uniformly classified the entire study area to generate annual land cover products.

However, this manuscript proposes a dynamic mapping method that updates annually, offering a novel approach to enhancing interannual stability. First, a reference vegetation map of the QTP for 2020 was produced using terrain-climate-remote sensing data. The CCD model was then applied to detect potential change areas, identifying the temporal and spatial positions of breakpoints. The RF model was used only for pixel-based classification in these breakpoint regions, while non-breakpoint areas were not reclassified. Finally, a spatio-temporal consistency approach was implemented to reduce misclassified pixels, enhancing interannual stability.

The key innovation of this study lies in dynamically updating the mapping annually, rather than classifying the entire study area each year. By reducing the number of pixels subject to classification, we minimize the risk of misclassification. Specifically, for non-breakpoint areas, labels were derived directly from the nearest neighbor vegetation map from the preceding year. For example, the non-breakpoint areas in 2019 were assigned labels based on the 2020 map. This approach significantly differs from the "RF+CCDC+Spatial-temporal Consistency" framework of Xian et al. (2021), as our method focuses on reducing the classification area and employing dynamic updates to ensure interannual consistency.

In summary, the methodological innovation of this study lies in its dynamic, annually updated mapping approach, which aims to address the challenge of interannual stability in land cover products.

*Q6:* Finally, the language issues. There are many subjective and vague descriptions lacking proper citations in the manuscript. For instance, Section 2.3.2: "Additionally, 6 percentiles: 5%, 30%, 45%, 60%, 75%, and 90%, were calculated for the 7 reflectance bands and the 14 indices. " How were these percentiles selected?

*Answer:* Thank you for your suggestion. We apologize for the error, and we have corrected "5%" to "15%" in the revised manuscript (see line 197). Additionally, we have included relevant explanations and citations to clarify the selection of these percentiles (see lines 198-202). In this study, six percentiles (15%, 30%, 45%, 60%, 75%, and 90%) were calculated from intra-annual observations of time series remote sensing imagery to create a vegetation mapping feature set. The 15% and 90% percentiles were chosen as substitutes for the minimum and maximum values to mitigate the influence of extreme values in the time series data (Zhang et al., 2021). Meanwhile, the intermediate percentiles (30%, 45%, 60%, and 75%) were selected to capture temporal trends evenly across the time series, ensuring uniformity without creating feature redundancy (Sulla-Menashe et al., 2019; Zhang et al., 2024).

*Q7:* Section 2.2.3: "For the QTP within China, climate data at 1,000 m was obtained from the National Tibetan Plateau Data Center. " What specific climate data was used?

*Answer:* Climate data for the QTP within China were obtained from the National Qinghai-Tibet Plateau Data Center, specifically the "1-km monthly precipitation dataset for China (1901-2023)" and the "1-km monthly mean temperature dataset for China (1901-2023)." To derive the precipitation and temperature features used in this study, the monthly precipitation data were summed to calculate annual totals, and the monthly mean temperature data were averaged to obtain annual mean temperatures. For relevant details on the climate data and their citation, please see the revised manuscript, lines 178-181.

*Q8:* The Discussion also repeatedly uses subjective phrases such as, "Fortunately, our annual vegetation map also reflected this characteristic, " and "Our long-time annual vegetation maps also reflected this characteristic, " which undermines the objectivity of the writing.

*Answer:* Thank you for your suggestion. We have revised the subjective language in the manuscript. For instance, we changed "Fortunately, our annual vegetation map also reflected this characteristic" to "The annual vegetation maps accurately captured this transition" and "Our long-time annual vegetation maps also reflected this characteristic" to "The long-time annual vegetation maps consistently reflected this characteristic." In addition, we reviewed the Discussion section and removed other subjective phrases to maintain objectivity. Please refer to Sections 4.1 and 4.2 of the revised manuscript.

**References**

Feng, M., Huang, C., Channan, S., Vermote, E. F., Masek, J. G., and Townshend, J. R.: Quality assessment of Landsat surface reflectance products using MODIS data, Comput. Geosci., 38, 9–22, https://doi.org/10.1016/j.cageo.2011.04.011, 2012.

Li, C., Peng, G., Wang, J., Zhu, Z., Biging, G. S., Yuan, C., Hu, T., Zhang, H., Wang, Q., and Li, X.: The first all-season sample set for mapping global land cover with Landsat-8 data, Sci. Bull., 62, 508–515, https://doi.org/10.1016/j.scib.2017.03.011, 2017.

Liu, L., Zhang, X., Gao, Y., Chen, X., Shuai, X., and Mi, J.: FinerResolution Mapping of Global Land Cover: Recent Developments, Consistency Analysis, and Prospects, J. Remote Sens., 2021, 5289697, https://doi.org/10.34133/2021/5289697, 2021.

Sulla-Menashe, D., Gray, J. M., Abercrombie, S. P., and Friedl, M. A.: Hierarchical mapping of annual global land cover 2001 to present: The MODIS Collection 6 Land Cover product, Remote Sens. Environ., 222, 183-194, https://doi.org/10.1016/j.rse.2018.12.013, 2019.

Xian, G. Z., Smith, K., Wellington, D., Horton, J., Zhou, Q., Li, C., Auch, R., Brown, J. F., Zhu, Z., and Reker, R. R.: Implementation of the CCDC algorithm to produce the LCMAP Collection 1.0 annual land surface change product, Earth Syst. Sci. Data, 14, 143–162, https://doi.org/10.5194/essd-14-143-2022, 2022.

Zhang, X., Liu, L., Chen, X., Gao, Y., Xie, S., and Mi, J.: GLC_FCS30: global land-cover product with fine classification system at 30m using time-series Landsat imagery, Earth Syst. Sci. Data, 13, 2753–2776, https://doi.org/10.5194/essd-13-27532021, 2021.

Zhang, X., Zhao, T., Xu, H., Liu, W., Wang, J., Chen, X., and Liu, L.: GLC_FCS30D: the first global 30 m land-cover dynamics monitoring product with a fine classification system for the period from 1985 to 2022 generated using dense-time-series Landsat imagery and the continuous change-detection method, Earth Syst. Sci. Data, 16, 1353–1381, https://doi.org/10.5194/essd-16-1353-2024, 2024.

**Thanks to the reviewer for these valuable feedback and valuable suggestions.**

---

## Author Response (AR3)

**Responses to Reviewers Comments**

**Reviewer #1**

After several revisions, the questions raised by the reviewer have been well addressed and improved, meeting the publication requirements of the journal.

**Thank you very much for the valuable feedback and suggestions.**

**Reviewer #2**

The manuscript combined MODIS data and random forest machine learning algorithms to map long-term vegetation on the Qinghai-Tibet Plateau, and considered the CCDC algorithm to reduce uncertainty on the time scale. The results could present data basis for understanding the vegetation response mechanism under the background of climate change. However, the introduction and method description of the present version of the manuscript are not convincing, making it difficult to assess the uncertainty of the results. Therefore, I recommend to reject at its present state. I list my major concerns as follows:

*Q1:* The abstract does not clearly specify the new method. Please include a brief description of the main aspects of the new method.

 *Answer:* Thank you for your suggestion. We have added a description of the key aspects of the new method in the abstract. Please see lines 15-17 in the revised manuscript.

*Q2:* The term "independent classification method" is not commonly used in remote sensing image classification studies. Could you consider using a more widely recognized term?

 *Answer:* Thank you. We have revised "independent classification methods for each period's product" to "traditional classification methods for each period's product." Please refer to line 13 in the revised manuscript.

*Q3:* Please distinguish the citation for Liu et al., 2021. There are two authors named Liu who published articles in 2021 in the reference list, and it is unclear which specific paper is being cited.

 *Answer:* Thank you. We have added "a" and "b" annotations to the two references by Liu et al., 2021 in the revised manuscript to distinguish between them. The specific changes can be found in lines 40 and 43 of the revised version.

*Q4:* The climate data sources within and outside of China are not unified, with one having a resolution of 1km and the other 50km, resulting in a significant difference. Downscaling is required to unify the resolution. However, the author did not specify the subsequent processing steps for the climate data.

 *Answer:* Thank you for your suggestion. In this study, we adopted a mean resampling method based on Google Earth Engine for data processing. All climate data, including 1 km resolution data in China and 50 km resolution data outside China, were resampled to a 500 m resolution to ensure consistency with other datasets, such as remote sensing imagery.

*Q5:* For each vegetation category, please provide descriptive information to help readers

understand their characteristics. For instance, alpine scrub meadow (ASM), alpine meadow (AM), alpine grassland (AG), and alpine vegetation (AV) may be difficult to distinguish by name alone. A table with descriptions for each category and the number of sample points would enhance the article's readability.

*Answer:* Thank you for your suggestion. In the revised manuscript, we have added Table 1, which includes the names, definitions, and the number of training and validation samples for each of the 16 vegetation types in this study. To further enhance clarity, the number of training and validation samples is also visually represented in a bar chart in Figure 1. Please refer to line 106 in the revised manuscript for these updates.

*Q6:* When selecting training samples for the classification of the 2020 MODIS dataset, based on the 10-meter resolution QTP vegetation classification map, why was 70% purity chosen? Is there any basis for 70% in other studies?

*Answer:* Thank you. In this study, a purity threshold of 70% was chosen as the criterion for training sample selection, primarily based on the characteristics of the study area and the quality requirements. In Table 2, we present the area proportions for different purity levels, with regions of greater than 70% purity covering approximately 62.34% of the area of the QTP (in China). Choosing a higher purity threshold would significantly reduce the available sampling area, which is not conducive to collect samples for smaller and more dispersed vegetation types. On the other hand, choosing a lower purity threshold would increase the proportion of mixed pixels, affecting the reliability of the samples. Taking these factors into account, we believe that a 70% purity threshold strikes a good balance between sample quality and the available sampling area in this study.

*Q7:* "Each sample was independently interpreted regarding its vegetation type by three interpreters, using long-time series temperature and precipitation data as well as Landsat remote sensing images." Could you clarify how the long-term temperature and precipitation data were used to determine the sample class?

*Answer:* Thank you for your question. The visual interpretation of vegetation types on the QTP primarily relies on long-term Landsat remote sensing imagery, with temperature and precipitation data serving as auxiliary information for sample quality control. Different vegetation types usually have specific climatic suitability ranges. For example, alpine meadows have a higher demand for precipitation and temperature, while alpine grasslands are adapted to drier and colder environments. By integrating long-term temperature and precipitation data, we can verify whether the climatic conditions of each sample are consistent with the characteristics

of its corresponding vegetation type. This allows us to exclude samples that significantly deviate from the suitable climatic range, thereby enhancing sample quality and ensuring the scientific accuracy of the classification results.

*Q8:* In Section 2.2.2, it might be helpful to summarize the main information and move the detailed methodology to supplementary information. This could improve the article's readability.

*Answer:* Thank you for your suggestion. After careful consideration, this section mainly describes the process of obtaining validation samples, which is closely related to the main content and presented in a textual format. Moving it to the supplementary information might compromise the reader's understanding of the sample acquisition process. To improve readability, we have streamlined this section by removing redundant descriptions while retaining the key information.

*Q9:* "This study utilized climate data which included annual precipitation (AP) and annual average temperature (AT) across the entire QTP from 2000 to 2022." Please clarify the specific role of the climate data to help readers understand its importance in the research.

*Answer:* Thank you. We have added content in Section 2.2.4 to emphasize the importance of climate data in determining vegetation distribution on the QTP. Please see lines 177-179 in the revised manuscript.

*Q10:* Please revise Equation 6 to avoid any misunderstanding that n can only go up to 3.

*Answer:* Thank you for your suggestion. We have revised the formula and explanatory text to avoid the misunderstanding that the upper limit of $n$ can only be 3. In the updated version, $N$ has been introduced in the formula to represent the maximum order of harmonics, indicating that this value can be adjusted flexibly according to research needs rather than being fixed at 3. Please see lines 259-261 in the revised manuscript.

*Q11:* The "Description" section of Table 2 requires more detail. Instead of abbreviations like MIR or SWIR1, please provide the full name and specify the wavelength details. Additionally, for the indices, adding a brief description of their characteristics, such as their physical significance, would be helpful.

*Answer:* Thank you. We have revised Table 3 (formerly Table 2), supplemented the full name and wavelength ranges of the bands, as well as annotations explaining the physical significance of each index along with the corresponding references. Please see line 208 in the revised manuscript.

*Q12:* "The 'breakpointBands' parameter specifies the bands for breakpoint detection, including Red, NIR, and SWIR 1, which correlate with chlorophyll content, leaf structure, and water content, respectively." Please ensure consistency with the abbreviations used earlier, particularly if SWIR 1 refers to S1 (Swir1).

*Answer:* Thanks. We have carefully reviewed and ensured consistency between the abbreviations used in the text and those in Table 3. For example, R, N, and S1 represent the Red, Near-infrared, and Shortwave Infrared 1 bands, respectively. Please see lines 276-277 in the revised manuscript.

*Q13:* "The vegetation map of the QTP includes 16 types (Fig. 4)." Please specify that this refers to the 2020 classification.

*Answer:* Thanks. We have revised the text to: "The 2020 vegetation map of the QTP includes 16 types (Fig. 4)." to clarify that it refers to the classification results for 2020. Please see line 323 in the revised manuscript.

*Q14:* In Section 4.1, it would be helpful to explain why the Red, NIR, and SWIR 1 bands were chosen for breakpoint detection in CCD. Could you discuss the relevance of these bands for breakpoint identification based on the characteristics of QTP vegetation?

*Answer:* Thank you. In Section 4.1, we have added the reasons for selecting the R, N, and S1 bands for breakpoint detection in the CCD process. Specifically, the R band is related to vegetation chlorophyll content, the N band is associated with vegetation canopy structure, and the S1 band reflects vegetation water content. Additionally, these three bands are commonly used to construct vegetation indices such as NDVI and SAVI. Please see lines 427-428 in the revised manuscript.

*Q15:* In the discussion, it would be useful to elaborate on why certain variables were more significant in the 2020 RF classification. Have these variables been found to be important in other studies as well? Is the QTP classification unique, making some variables more critical?

*Answer:* Thank you for your suggestion. In previous study on the 10-meter vegetation map of the QTP, we have demonstrated the importance of elevation, annual average temperature (AT), annual precipitation (AP), slope, and aspect in the 2020 RF classification (Zhou et al., 2023). However, studies focusing on vegetation distribution at the 500-meter scale for this region are relatively scarce. At this scale, we found that elevation, AT, AP, and slope remain key variables, while the influence of aspect diminishes. Given the limited impact of human activities on the QTP (Chen et al., 2023), the selected variables in this study further highlight the unique geographical and climatic conditions that shape vegetation classification in this

region.

*Q16:* Lines 29-30: Among ESA's land cover data products, ESA-CCI is more representative because it has good temporal continuity and a spatial resolution of 300 m.

 *Answer:* Thank you for your suggestion. We have explicitly introduced the ESA-CCI land cover data product in the manuscript and highlighted its provision of excellent temporal continuity at a 300 m spatial resolution. Please see lines 30-31 in the revised manuscript.

*Q17:* Lines 41-43: Why can't direct comparisons be made if there are significant differences? How do the authors rule out situations where these differences actually occur. As far as I know, the mainstream data products in the world are obtained by classifying single-period data after pre-training.

 *Answer:* Thank you for your question. The differences in data processing and classification methods between GlobCover 2005 and 2009 make these maps more suitable as independent land cover datasets rather than as a basis for change detection (Bontemps et al., 2010). Liu et al. (2021) pointed out that single-period products from different years often suffer from inconsistencies in spatial and temporal accuracy due to variations in classification algorithms or data processing, which can lead to misinterpretation of processing noise as actual changes. Even high-resolution (30 m) GLC products have significant classification errors in complex terrains, such as shrublands and grasslands, which limit their capability for interannual change analysis. In contrast, dynamic change detection methods, such as Continuous Change Detection and Classification (CCDC), enhance interannual stability through time-series analysis and differentiate between classification errors and true changes. For instance, the GLC_FCS30D product (Zhang et al., 2024b) used the CCDC method to generate a global 30-meter dynamic land cover dataset for 1985–2022, effectively improving temporal consistency. This demonstrates that global land cover monitoring has gradually shifted from single-period classifications to time-series analyses, which significantly improving monitoring accuracy and interannual stability.

*Q18:* Lines 39-54: The background introduction of the current version is not convincing. Will the data products that take into account time continuity bring uncertainty to the mapping results of areas where vegetation mutations occur on the Qinghai-Tibet Plateau?

 *Answer:* Thank you. According to your suggestion, we have revised the background section to further emphasize the issues of spatial and temporal consistency in traditional long-term data products, and clarified the solutions based on post-processing methods and continuous change detection techniques. In particular, the CCDC method significantly enhances the stability of

time-continuous data products, especially in areas with large changes. This method effectively identifies and reduces the temporal instability of time-series classification products. Please see lines 41-59 in the revised manuscript.

*Q19:* Lines 55-60: Are these all the author's opinions? Why are there no references?

   *Answer:* Thank you for your comments. This section is based on conclusions supported by existing literature. In the revised manuscript, we have added the corresponding references to more accurately reflect the sources of this information. Please see lines 60-66 in the revised version.

*Q20:* Lines 61-72: There is insufficient summary of vegetation mapping research at the scale of the Qinghai-Tibet Plateau, and there is a lack of summaries of some influential studies. For example, https://doi.org/10.1007/s11629-015-3485-y.

   *Answer:* Thanks. We have added a summary of vegetation mapping studies at the QTP, including the influential study you mentioned (DOI: 10.1007/s11629-015-3485-y). Additionally, we have referenced other relevant studies (DOI: 10.1016/j.ecolind.2022.108599; DOI: 10.1016/j.scib.2023.07.035) to provide a more comprehensive overview. Please see lines 71-76 in the revised manuscript for the specific changes.

*Q21:* Lines 86-87: What is the "permanent glaciers"? Glaciers, snow cover and permafrost are not vegetation.

   *Answer:* Thank you. We have removed the sections on "permanent glaciers, snow cover, and permafrost," which are unrelated to vegetation, to maintain accuracy and relevance to vegetation distribution. Please see lines 94-96 in the revised manuscript.

*Q22:* Lines 90-95: This classification strategy is also designed from the perspective of land cover. How did the authors start from the perspective of vegetation cover? In addition, what strategy did the authors use to distinguish "evergreen broad-leaved forest (EBF), evergreen coniferous forest (ECF), coniferous and broad-leaved mixed forest (CBMF), deciduous broad-leaved forest (DBF), and deciduous coniferous forest (DCF)" at a spatial resolution of 500 m? Are they clearly distinguishable at such a coarse spatial resolution? Has their spectral separability been evaluated? This is my main concern. In addition, how are "alpine vegetation (AV) and cultivated vegetation (CV)" defined, and how are they distinguished from other vegetation? Does the bare land type on the Qinghai-Tibet Plateau only include "alpine desert (AD)"?

*Answer:* Thank you. The classification strategy in this study focuses on vegetation cover, integrating ground survey data and spectral validation to ensure that different vegetation types (e.g., EBF, ECF, CBMF, DBF, DCF) can be reasonably distinguished at a 500 m resolution. While the resolution is relatively coarse, extensive ground surveys and spectral analyses conducted by our team in the QTP indicate that these vegetation types have separability in spectral characteristics and ecological distributions, allowing effective differentiation using a random forest classifier. It is important to note that, unlike land cover products such as GlobeLand30 (Chen et al., 2015), which classify categories like forest and grassland, this study focuses on a more refined delineation of vegetation types, such as alpine meadow, alpine grassland, and alpine desert. To aid readers' understanding of the classification criteria, we have added definitions and characteristic descriptions of "alpine vegetation (AV) and cultivated vegetation (CV)" in Table 1. Regarding desert types, the "alpine Desert (AD)" in this study is defined based on the Editorial Board of the Vegetation Map of China, Chinese Academy of Sciences (2007), which is characterized by the Deserts composed of cold- and drought-tolerant cushion subshrubs.

*Q23:* Lines 96-111: I have the same concerns as the first round of reviewers that the errors of the 10 m resolution data product will be introduced into the results of this study. In addition, the authors did not avoid the existence of mixed pixels during the upscaling process, which is another concern of mine, and will bring great uncertainty to the results. Homogeneous areas should be selected to form training data, and I am not sure whether 70% purity is sufficient.

*Answer:* Thanks. While the 10 m resolution vegetation map serves as the reference data for this study, its influence on the results has been managed. We selected training samples using a 70% purity threshold and validated the results with independent third-party samples. Additionally, mixed pixels are indeed a challenge at the 500 m scale, but we have mitigated this issue by selecting training samples only from areas with a purity greater than 70%. This threshold ensures that the dominant vegetation type constitutes the majority of the pixel, enhancing the representativeness of the training data. A higher threshold, while potentially reducing uncertainties, would significantly reduce the number of samples, particularly for less common vegetation types. Conversely, a lower threshold would increase the proportion of mixed pixels, diminishing the quality of the training data. The choice of 70% purity represents a balance between sample quality and availability, as approximately 62.34% of the QTP (within China) consists of areas exceeding this purity level, ensuring sufficient high-quality samples for robust model training. In summary, this study has minimized the error propagation from the 10-meter vegetation map as much as possible. At the same time, the 70% purity threshold

provides an effective balance between ensuring sample quality and maintaining sufficient sample coverage.

*Q24:* Lines 105-107: Why is this method not used in China? Generally speaking, the sample data obtained by visual interpretation is more accurate. Is it sufficient to use only one year of sample data to generate classification results for such a long time series?

*Answer:* Thank you for your question. We used the " Vegetation map of Qinghai Tibet Plateau in 2020 with 10 m spatial resolution" as the basis for training samples due to its high resolution and reliability. The vegetation map was generated at a 10-meter spatial resolution with an overall accuracy of 89.5%, constructed based on visual interpretation and multi-source data integration (terrain, climate, and remote sensing information) (Zhou et al., 2023). Such high-precision foundational data provides support for vegetation classification at the 500-meter scale. Additionally, as vegetation types on the QTP are primarily influenced by climate, with less human activity impact (Chen et al., 2023), it is reasonable to use 2020 sample data to generate long-term classification results. Our methodology not only ensures consistency of the classification system but also incorporates change detection techniques, enabling the capture of dynamic vegetation changes over time.

*Q25:* Section 2.2.2: The selection method of validation samples is more standardized than that of training samples. The method of selecting validation samples is more standardized than that of training samples. Why don't we follow the standards when selecting training samples?

*Answer:* Thank you. The selection methods for training and validation samples differ due to their distinct purposes: validation samples aim to strictly control quality to assess model accuracy, and thus a more rigorous filtering approach is adopted. Training samples focus on quantity and diversity to meet the model's learning requirements. The training samples were primarily derived from a high-resolution vegetation map with an overall accuracy of 89.5%. To ensure their representativeness and consistency, samples were filtered based on pixel purity (≥70%). Additionally, high-resolution imagery interpretation was integrated to ensure adequate spatial and type coverage. Although the validation sample standards were not fully adopted for training, the current approach achieves a balance between sample quantity and quality, effectively supporting model training.

*Q26:* Section 2.2.3: The introduction of preprocessing is too brief. The pretreatment methods and standards used need to be introduced, and relevant literature should be cited.

*Answer:* Thank you for your suggestion. In Section 2.2.3, we have added a detailed description of the preprocessing methods for MOD09A1 data, including its generation as a

standardized NASA Level 3 data product and the additional processing steps we applied in this study, such as coordinate system transformation and spatial clipping. Relevant references have also been added. Please see lines 165-176 in the revised manuscript.

*Q27:* Lines 176-178: How to draw this conclusion from this reference?

   *Answer:* Thank you for your valuable comments. The climate data used in this study were obtained from publicly available datasets provided by the National Tibetan Plateau Data Center, namely the "1-km Monthly Precipitation Dataset for China (1901–2023)" and the "1-km Monthly Mean Temperature Dataset for China (1901–2023)." These references are relevant literature explicitly requested by the data provider to be cited, aiming to acknowledge the source and generation method of the data and ensure its reliability and scientific validity. Therefore, these citations are included to comply with data usage specifications rather than directly supporting the conclusions of this study.

*Q28:* Lines 185-188: Direct resampling is no different from using the original spatial resolution and may even reduce product accuracy, especially from 50000m to 500m.

   *Answer:* Thank you for your insightful comments. In the mapping process of this study, in addition to temperature and precipitation data, vegetation indices derived from MODIS data and topographic features played a significant role. Although meteorological data with a 50 km spatial resolution were resampled directly to 500 m, the inherent 500 m resolution of MODIS data and the 30 m resolution of topographic data helped mitigate potential impacts on product accuracy. Additionally, we chose two regions for validation. Cultivated vegetation shown in Fig. S1a is clearly represented in Fig. S1b, while water bodies, glaciers, and snow in Fig. S1c are accurately depicted in Fig. S1d. It is worth noting that the display areas in Fig. S1 are all within 50 km. Therefore, it is feasible to use 50 km resolution meteorological data for regions outside China, and high-accuracy 500 m classification results can be obtained.

[Figure]

**Fig. S1 Comparison of 2020 Landsat imagery and classification results for regions outside China on the QTP**

*Q29:* Section 2.3.2: Why were these features chosen?

*Answer:* Thank you. We selected these features based on the following considerations: first, vegetation indices (e.g., NDVI, EVI, etc.) and spectral characteristics (e.g., red, near-infrared bands, etc.) are core indicators widely used in vegetation classification and monitoring in existing literature. They can effectively reflect the physiological condition and spectral properties of vegetation. Second, topographic features (e.g., elevation, slope, etc.) are critical in the QTP region, as vegetation distribution in this region is significantly related to elevation and terrain. Additionally, meteorological data (e.g., AT and AP) are key drivers of vegetation distribution. Therefore, the selection of these features is based on the theoretical support from previous research and the understanding of the environmental characteristics of the study area (Wang et al., 2022, Wang et al., 2023; Zhang et al., 2024a), which can effectively enhance the scientific validity and accuracy of the classification.

*Q30:* Section 2.3.3: I could not understand from the current introduction and the cited reference the method adopted to evaluate feature importance, nor could I assess whether it could be used to rule out multicollinearity.

*Answer:* Thank you. We have provided a detailed description of the feature importance evaluation method and the means of dealing with multicollinearity in the manuscript. Specifically, to address potential multicollinearity issues in optical features, we introduced the Variance Inflation Factor (VIF) method. This allowed us to quantitatively filter out features with severe collinearity, thereby reducing model redundancy and computational load. Subsequently, by combining the feature importance ranking with the VIF selection results, optimal feature combination was selected for vegetation classification on the QTP based on the minimal out-of-bag error. The methods employed in this study are widely used in relevant literature and have been verified for their reliability and applicability in experiments (Ngabire et al., 2022; Zhang et al., 2019).

*Q31:* Section 2.4.1: The necessity and advantages of using this algorithm in this study should be mentioned.

*Answer:* Thanks for your suggestion. In section 2.4.1, we have added an explanation of the necessity and advantages of using the CCDC algorithm in this study. Specifically, we describe that it improves the temporal consistency of the classification products by considering both temporal and spectral features to identify the "breakpoint" areas and time periods of vegetation on the QTP from 2000 to 2022. Please see the revised manuscript, lines 242-247.

*Q32:* Section 2.4.3: References?

*Answer:* Thank you for your suggestion! The method described in section 2.4.3 is an

innovative approach we proposed specifically for the data characteristics and research needs of this study. The detailed operational process is clearly presented through formulas, and as such, this section does not reference any external literature.

*Q33:* Lines 298-299: How do authors consider local-scale transient disturbances induced by human activities or extreme climate?

*Answer:* Thank you for your question. Intense human activities and extreme climate events may indeed cause transient disturbances at the local scale on the Qinghai-Tibet Plateau. However, a study by Chen et al. (2023) points out that the impact of human activities on the plateau is relatively weak, and its growth rate has slowed since 2010. Furthermore, despite the fact that the warming rate on the QTP is twice the global average, the region has generally experienced a greening trend, with ecological recovery and overall good environmental quality (Chen et al., 2023). Additionally, considering that the CCDC algorithm used in this study has strong change detection capabilities (Pasquarella et al., 2022), it is able to effectively capture and detect any transient disturbances caused by extreme climate events or human activities on the plateau.

**References**

Bontemps, S., Defourny, P., Bogaert, E. V., Arino, O., Kalogirou, V., and Perez, J. R.: GLOBCOVER 2009 Products Description and Validation Report, available at: http://due.esrin.esa.int/files/GLOBCOVER2009_Validation_Report_2.2.pdf (last access: 29 September 2024), 2010.

Chen, F. H., Zhang, Y. L., Hou, J. Z., and Fan, J. W.: Scientific Assessment of Human Activities on the Ecological Environment of the Qinghai-Tibet Plateau, China Meteorological Press, Beijing, China, ISBN 9787502979515, 2023.

Chen, J., Chen, J., Liao, A., Cao, X., Chen, L., Chen, X., He, C., Han, G., Peng, S., Lu, M., Zhang, W., Tong, X., and Mills, J.: Global land cover mapping at 30 m resolution: A POK-based operational approach, ISPRS J. Photogramm. Remote Sens., 103, 7-27, https://doi.org/10.1016/j.isprsjprs.2014.09.002, 2015.

Liu, L., Zhang, X., Gao, Y., Chen, X., Shuai, X., and Mi, J.: FinerResolution Mapping of Global Land Cover: Recent Developments, Consistency Analysis, and Prospects, J. Remote Sens., 2021, 5289697, https://doi.org/10.34133/2021/5289697, 2021.

Ngabire, M., Wang, T., Xue, X., Liao, J., Sahbeni, G., Huang, C., Duan, H., and Song, X.: Soil salinization mapping across different sandy land-cover types in the Shiyang River Basin: A remote sensing and multiple linear regression approach, Remote Sens. Appl. Soc. Environ., 28, 100847, https://doi.org/10.1016/j.rsase.2022.100847, 2022.

Pasquarella, V. J., Arévalo, P., Bratley, K. H., Bullock, E. L., Gorelick, N., Yang, Z., and Kennedy, R. E.: Demystifying LandTrendr and CCDC temporal segmentation, Int. J. Appl. Earth Obs., 110, 102806, https://doi.org/10.1016/j.jag.2022.102806, 2022.

Wang, Y., Sun, J., He, W., Ye, C., Liu, B., Chen, Y., Zeng, T., Ma, S., Gan, X., Miao, C., Zhou, H., and Tsunekawa, A.: Migration of vegetation boundary between alpine steppe and meadow on a century-scale across the Tibetan Plateau, Ecol. Indic., 136, 108599, https://doi.org/10.1016/j.ecolind.2022.108599, 2022.

Wang, Y., Xue, K., Hu, R., Ding, B., Zeng, H., Li, R., Xu, B., Pang, Z., Song, X., Li, C., Du, J., Yang, X., Zhang, Z., Hao, Y., Cui, X., Guo, K., Gao, Q., Zhang, Y., Zhu, J., Sun, J., Li, Y., Jiang, L., Zhou, H., Luo, C., Zhang, Z., Gao, Q., Chen, S., Ji, B., Xu, X., Chen, H., Li, Q., Zhao, L., Xu, S., Liu, Y., Hu, L., Wu, J., Yang, Q., Dong, S., He, J., Zhao, X., Wang, S., Piao, S., Yu, G., and Fu, B.: Vegetation structural shift tells environmental changes on the Tibetan Plateau over 40 years, Sci. Bull., 68, 1928 – 1937, https://doi.org/10.1016/j.scib.2023.07.035, 2023.

Zhang, J., Okin, G. S., and Zhou, B.: Assimilating optical satellite remote sensing images and field data to predict surface indicators in the Western US: Assessing error in satellite predictions based on large geographical datasets with the use of machine learning, Remote Sens. Environ., 233, 111382, https://doi.org/10.1016/j.rse.2019.111382, 2019.

Zhang, L., Shen, M., Yang, Z., Wang, Y., and Chen, J.: Spatial variations in the difference in elevational shifts between greenness and temperature isolines across the Tibetan Plateau grasslands under warming, Sci. Total Environ., 906, 167715, https://doi.org/10.1016/j.scitotenv.2023.167715, 2024a.

Zhang, X., Zhao, T., Xu, H., Liu, W., Wang, J., Chen, X., and Liu, L.: GLC_FCS30D: the first global 30 m land-cover dynamics monitoring product with a fine classification system for the period from 1985 to 2022 generated using dense-time-series Landsat imagery and the continuous change-detection method, Earth Syst. Sci. Data, 16, 1353 – 1381, https://doi.org/10.5194/essd-16-1353-2024, 2024b.

Zhou, G., Ren, H., Liu, T., Zhou, L., Ji, Y., Song, X., and Lv, X.: A new regional vegetation mapping method based on terrain-climate-remote sensing and its application on the Qinghai-Xizang Plateau, Sci. China Earth Sci., 66, 237-246, https://doi.org/10.1007/s11430-022-1006-1, 2023.

**Thank the reviewer very much for these constructive comments and suggestions.**

---

## Author Response (AR4)

**Responses to Reviewers Comments**

**Reviewer #6**

I think this manuscript has been revised in accordance with the review comments and is ready for publication.

**Thank you very much for the valuable feedback and suggestions.**

**Reviewer #7**

I am glad to read the ms of RS-based vegetation mapping in Tibetan Plateau, where the regional in-situ observation is impossible. Generally, I thought this ms an interesting work and the useful result derived. I read through the answers from authors to previous reviewing comments, and also though several issues to be re-considered before publication.

***Q1:*** Method of this dataset was based on detecting actual changes in vegetation covers between two neighbouring years. However, the total change in vegetation types at 500-m scale between two years is not highly possible. Although the samples with high pixel purity were chosen in RF training, the authors should make sure of the pixels with potential changes in vegetation covers meeting the purity requirement (so that the pre-trained RF model could be used).

*Answer:* Thank you for your suggestion. As you pointed out, the likelihood of significant changes in vegetation types between two years at a 500 m resolution is relatively low. However, our approach ensures that both the selected training samples and the pixels with potential vegetation changes meet purity criteria. Mixed pixels are common in remote sensing applications, especially at lower resolutions. According to remote sensing theory, the dominant type of a pixel (i.e., the type with the largest proportion) typically represents the pixel's overall attribute. Therefore, when selecting training samples, we choose high-purity pixels to ensure that these samples reflect the vegetation type they represent. It is worth noting that changes in the minor components of a pixel typically do not have a significant impact on the pixel's overall attribute and are challenging to detect as potential changes at 500 m resolution. Consequently, our change detection algorithm focuses on identifying significant changes in the dominant type, ensuring that the detected change areas maintain high purity and meet the analytical requirements of the study.

***Q2:*** The RF model was trained in 2020, and the result of 2020 was then used to update the maps of rest years gradually. The errors in 2020 would propagate to other year. Did authors compare the maps of rest years directly predicted using 2020-trained RF model with the maps of current gradually updating method?

*Answer:* Thank you for your comments. Due to the temporal and spatial distribution differences of remote sensing imagery and the inherent randomness of machine learning classifiers, directly using the model trained on 2020 data to predict maps for other years may result in discontinuities and unreasonable changes (Liu et al., 2021; Du et al., 2023). We initially attempted to use the 2020 model to directly predict maps of other years on the QTP, but the results showed significant error accumulation. To address this issue, we adopted the gradually updating method, which has been demonstrated to be effective in previous studies. For example, Zhang et al. (2024) used a CCD-

based gradually updating method to generate GLC-FCS30D products with higher temporal and spatial consistency.

***Q3:*** Over past 20+ years, there were possible radiative drifts in the MODIS equipment, and could those influence identifying gradual transmission between vegetation types?

***Answer:*** Thank you for your suggestion. Over the past two decades, MODIS sensors have indeed experienced some degree of radiometric drift. However, the MODIS Calibration Team has significantly mitigated this issue in the Collection 6 (C6) products by implementing recalibrations involving the solar diffuser, lunar measurements, and scan angle corrections (Doelling et al., 2015). Studies have demonstrated that the radiometric stability of C6 products is maintained within 1% (Doelling et al., 2015). The MOD09A1 product utilized in this study is derived from the C6 collection, effectively minimizing the influence of radiometric drift on data quality. Furthermore, vegetation type classification in this study is not solely dependent on spectral information; it also integrates meteorological and topographic factors. As a result, the potential impact of radiometric drift on identifying gradual transitions between vegetation types is negligible.

**References**

Doelling, D. R., Wu, A., Xiong, X., and Gopalan, A.: The Radiometric Stability and Scaling of Collection 6 Terra- and Aqua-MODIS VIS, NIR, and SWIR Spectral Bands, IEEE Trans. Geosci. Remote Sens., 53, 4520–4535, https://doi.org/10.1109/TGRS.2015.2400928, 2015.

Du, Z., Yu, L., Li, X., Zhao, J., Chen, X., Xu, Y., Yang, P., Yang, J., Peng, D., Xue, Y., and Gong, P.: Integrating remote sensing temporal trajectory and survey statistics to update land use/land cover maps, Int. J. Digit. Earth, 16, 4428-4445, https://doi.org/10.1080/17538947.2023.2274422, 2023.

Liu, L., Zhang, X., Gao, Y., Chen, X., Shuai, X., and Mi, J.: FinerResolution Mapping of Global Land Cover: Recent Developments, Consistency Analysis, and Prospects, J. Remote Sens., 2021, 5289697, https://doi.org/10.34133/2021/5289697, 2021.

Zhang, X., Zhao, T., Xu, H., Liu, W., Wang, J., Chen, X., and Liu, L.: GLC_FCS30D: the first global 30 m land-cover dynamics monitoring product with a fine classification system for the period from 1985 to 2022 generated using dense-time-series Landsat imagery and the continuous change-detection method, Earth Syst. Sci. Data, 16, 1353–1381, https://doi.org/10.5194/essd-16-1353-2024, 2024.

**Thanks to the reviewer for these valuable feedback and valuable suggestions.**

**Reviewer #8**

The authors have generated a 500-meter resolution vegetation distribution product for the Qinghai-Tibet Plateau from 2000 to 2022. The dataset was based on a previously created 10-meter resolution Vegetation map of the Qinghai-Tibet Plateau (2020), a series of spectral indices calculated from MOD09A1 products, and continuous change detection methods. The topic is interesting, and the product provides a new data source for the Qinghai-Tibet Plateau. However, I have some concerns about the expression and the innovation of the methods used in the paper:

*Q1:* I understand that the main significance of the paper is producing a new product rather than developing a new method. Currently, the authors emphasize the innovation of their methods too much, but the various methods used are not newly proposed in this paper; they have just not been applied to the Qinghai-Tibet Plateau before. I suggest that the authors focus more on data innovation and downplay the method innovation.

   *Answer:* Thank you for your suggestion. The core significance of this study lies in the innovation of the data product, rather than the innovation of the methodology. Based on your suggestion, we have simplified the description of the methods section to further highlight the uniqueness and scientific value of the data product. Please see lines 232-303 in revised manuscript.

*Q2:* The authors selected some pure pixels from the 10-meter resolution vegetation map as training data for the 500-meter resolution mapping. I have some questions about this. Although 10-meter resolution data have higher spatial resolution, this does not necessarily mean higher accuracy. In fact, the spatial and temporal consistency of high-resolution satellite reflectance is not as good as that of moderate-resolution satellites like MODIS, leading to more uncertainties. When creating the 10-meter vegetation map, the authors should have used local field sampling data. Why not use this data as training data?

   *Answer:* Thank you for your suggestion. To produce the 10 m resolution vegetation map of the Qinghai-Tibet Plateau (QTP), we utilized a limited but highly reliable set of field survey samples. To address the limited availability of high-quality samples, additional training samples were generated through visual interpretation in areas near existing samples, enabling the creation of the 10 m vegetation map. However, most field samples were collected in easily accessible areas such as roads and towns, where mixed pixel effects at the 500 m scale were significant, reducing the representativeness of these samples. Furthermore, only a small portion of field samples collected for validation through literature review passed the 500 m scale homogeneity filtering process. As a result, the field samples were not only limited in number but also exhibited low reliability at the 500 m scale, making them unsuitable as training data for this study. Given these limitations, we did not

use field samples as training data for vegetation mapping. Instead, we automatically generated highly representative samples in high-purity areas based on the existing 10 m vegetation map, allowing for the successful production of the 500 m resolution vegetation map of the QTP.

*Q3:* The authors used continuous temporal information to improve the mapping stability and introduced continuous change detection methods, which is very interesting. However, the description of the mapping methods and processes needs improvement. Figure 2 is an important figure showing the mapping process, but it is not intuitive enough. The figure labels subfigure numbers but lacks explanations. Moreover, the figure is difficult for readers to understand. I suggest improving it. Generally, figures should be self-explanatory and understandable without relying heavily on the text.

*Answer:* Thank you for your suggestion. We appreciate your recognition of the innovative aspects of our method, particularly the use of continuous temporal information and change detection to enhance mapping stability. Regarding Figure 2, we acknowledge that the original version may not have been sufficiently intuitive or self-explanatory. To address this, we have restructured the figure to present the mapping process in a more logical and visually engaging manner, reducing reliance on accompanying text. Please see lines 233-242 in the revised manuscript.

*Q4:* In Line 311, the text mentions "NDGlaI 15%, NDGlaI 90%, IBI 30%, NDBI 90%, IBI 15%, IBI 90%, EVI 90%, NDVI 90%, IBI 75%, IBI 60%, LSWI 90%, M 90%, and IBI 45%". What do these terms mean?

*Answer:* Thank you for your question. The terms mentioned in the text, such as "NDGlaI 15%" and "NDVI 90%", refer to the corresponding percentile values of the respective indices. For example, "NDGlaI 15%" indicates the value of NDGlaI at the 15th percentile of the sample data, while "NDVI 90%" refers the value of NDVI at the 90th percentile. Using these percentile values helps to better capture the distribution characteristics of each index, highlighting local value features of the variables, thereby improving classification accuracy in subsequent models.

*Q5:* The abbreviations in the text need to be standardized. Many abbreviations are uncommon or unnecessary. For example, LCSV is not a common abbreviation and is unnecessary, and its full forms are inconsistent in the text. Abbreviations like ECF are not necessary and are hard to understand. The full forms of methods like RF are missing. I suggest reviewing all abbreviations in the text and removing unnecessary ones.

*Answer:* Thank you for your suggestion. We have carefully reviewed all abbreviations in the text and adjusted those that are less commonly used or unnecessary. For instance, in the revised manuscript, we have added the full form of "RF" as "random forest" in lines 193-194 and updated

"LCSV" to its full form, "land cover and surface vegetation," in lines 12, 29, and 41. However, abbreviations such as "EBF" and "ECF" were retained to represent the 16 vegetation types on the QTP. While these abbreviations may not be universally familiar, their full forms are lengthy and occur frequently throughout the text and figures. To enhance the manuscript's readability, we decide to retain these abbreviations while providing their full forms and explanations upon their first mention. Furthermore, the use of 'EBF' for 'Evergreen Broadleaved Forest' aligns with the naming conventions frequently adopted in related studies (Zhang et al., 2024).

**References**

Zhang, X., Zhao, T., Xu, H., Liu, W., Wang, J., Chen, X., and Liu, L.: GLC_FCS30D: the first global 30 m land-cover dynamics monitoring product with a fine classification system for the period from 1985 to 2022 generated using dense-time-series Landsat imagery and the continuous change-detection method, Earth Syst. Sci. Data, 16, 1353–1381, https://doi.org/10.5194/essd-16-1353-2024, 2024.

**Thank the reviewer very much for these constructive comments and suggestions.**